# Unsupervised Representation Learning - an Invariant Risk Minimization Perspective

**Yotam Norman,  Ron Meir**
Department of Electrical & Computer Engineering
Technion - Israel Institute of Technology
yotamnor@gmail.com

## Abstract

We propose a novel unsupervised framework for *Invariant Risk Minimization* (IRM), extending the concept of invariance to settings where labels are unavailable. Traditional IRM methods rely on labeled data to learn representations that are robust to distributional shifts across environments. In contrast, our approach redefines invariance through feature distribution alignment, enabling robust representation learning from unlabeled data. We introduce two methods within this framework: Principal Invariant Component Analysis (PICA), a linear method that extracts invariant directions under Gaussian assumptions, and Variational Invariant Autoencoder (VIAE), a deep generative model that separates environment-invariant and environment-dependent latent factors. Our approach is based on a novel "unsupervised" structural causal model and supports environment-conditioned sample-generation and intervention. Empirical evaluations on synthetic dataset, modified versions of MNIST, and CelebA demonstrate the effectiveness of our methods in capturing invariant structure, preserving relevant information, and generalizing across environments without access to labels.

## 1 Introduction

Invariant Risk Minimization (IRM), introduced by Arjovsky et al. (2019), addresses the challenge of learning robust models in the presence of distribution shifts across environments (domains). This framework forms a cornerstone of the broader field of Out-Of-Distribution (OOD) learning, which seeks to enable models to generalize effectively to unseen environments. IRM considers scenarios with accessible training environments and inaccessible test environments, assuming that the underlying data distribution changes across environments. Within this setting, certain latent features of the data remain stable across environments (invariant features), while others change and are referred to as environmental or spurious features. The goal in IRM is to learn a representation of the data that preserves invariant features while filtering out environmental ones. A predictor or classifier trained on this representation is designed to exhibit robustness to distribution shifts, including shifts occurring in unseen environments that share the same underlying generative process. This work extends the concept of invariance in IRM to unsupervised settings, removing the dependency on labels or target values. By redefining invariance in the unsupervised context, we unlock new opportunities for research in IRM. Specifically, we propose methods to explore unsupervised algorithms within the IRM framework and demonstrate the feasibility of achieving invariant representations for unlabeled data. We present two novel approaches: Principal Invariant Component Analysis (PICA) and Variational Invariant Autoencoder (VIAE). PICA, grounded in Gaussian and linearity assumptions common in PCA literature, identifies a linear transformation that achieves invariant projections of the data. This dimensionality reduction method effectively filters out dimensions subject to distributional shifts while retaining invariant ones. VIAE, a variational autoencoder adapted for unsupervised IRM, is aimed at separating the latent space into invariant and environmental parts, allowing causal interventions for both generated samples and data points. This method is empirically tested on datasets inspired by common benchmarks in IRM literature (Gulrajani & Lopez-Paz, 2020), modified to suit the unsupervised framework. These contributions pave the way for new explorations in invariant representation learning, offering tools to handle distribution shifts in scenarios where labeled data is unavailable and/or expensive.

## 1.1 DEFINITIONS & FORMALIZATION

We consider a set of environments $\mathcal{E}_{\text{all}}$, each inducing a distinct probability distribution over the data: $X_e \sim P^e$ for each $e \in \mathcal{E}_{\text{all}}$. The set $\mathcal{E}_{\text{all}}$ is partitioned into two disjoint subsets: $\mathcal{E}_{\text{train}}$, containing the environments observed during training, and $\mathcal{E}_{\text{test}}$, containing environments that are not available during training. At inference time, samples may originate from either $\mathcal{E}_{\text{train}}$ or $\mathcal{E}_{\text{test}}$, depending on the context and the application.

Random vectors are denoted using uppercase letters (e.g., $Z_{\text{inv}}$, $Z_e$), where the subscripts $\text{inv}$ and $e$ indicate whether the variable is invariant to the environment or environment-dependent, respectively. Deterministic variables are denoted using lowercase letters. When a deterministic variable is written as a function of $e$, it implies direct dependence on the environment, i.e., $f(e)$ may vary across different environments $e$.

## 1.2 SUPERVISED AND UNSUPERVISED IRM

In the supervised IRM setup, the data consists of triplets $(X, Y, e)$, where $X$ denotes the input vector, $Y$ is the label, and $e$ specifies the environment. The data are assumed to be generated according to

$$(X, Y)_e \sim P^e_{X,Y}(x, y),$$

where $e$ is a known deterministic parameter that modulates the joint distribution over inputs and labels $(X, Y)$. The IRM optimization objective, originally formulated by Arjovsky et al. (2019), is given by:

$$\min_{\substack{\phi:\mathcal{X} \to \mathcal{H} \\ w:\mathcal{H} \to \mathcal{Y}}} \sum_{e \in \mathcal{E}_{\text{train}}} R^e(w \circ \phi) \quad ; \quad \text{s.t.} \quad w \in \underset{\bar{w}:\mathcal{H} \to \mathcal{Y}}{\arg\min} \ R^e(\bar{w} \circ \phi) \quad \forall e \in \mathcal{E}_{\text{train}}. \tag{1}$$

Where $R^e(\cdot)$ denotes the risk under environment $e$. Unlike standard empirical risk minimization (ERM), IRM introduces an additional constraint: the learned predictor $w \circ \phi$ must be optimal for each environment independently. Subsequent work (e.g. Zhou et al. (2022); Lin et al. (2022); Ahuja et al. (2021); Chen et al. (2022)) has focused on proposing surrogate objectives and algorithms to approximate solutions to this challenging bi-level optimization problem.

In this work, we extend the IRM framework to the unsupervised setting. Specifically, we investigate whether it is possible to learn invariant representations from unlabeled data drawn from multiple environments

$$X_e \sim P^e_X(x),$$

where the goal is to learn a feature map $\phi(X)$ such that

$$P^{e_1}(\phi(X)) = P^{e_2}(\phi(X)) \quad \forall e_1, e_2 \in \mathcal{E}_{\text{all}}.$$

In other words, the learned representation should be invariant to the environment. To this end, we consider a generative model parameterized by $\theta$, with environment-specific likelihoods $P^e_\theta(X)$. Our proposed objective is to maximize the sum of log-likelihoods across all environments, subject to the constraint that the induced distribution of the learned features $\phi(X)$ (also parametrized by $\theta$) is identical across environments. This leads to the unsupervised IRM optimization problem:

$$\max_\theta \sum_{e \in \mathcal{E}_{\text{train}}} \log P^e_\theta(X|\phi(X)) P^e_\theta(\phi(X)) \quad ; \quad \text{s.t.} \quad P^i_\theta(\phi(X)) = P^j_\theta(\phi(X)) \ \forall i, j \in \mathcal{E}_{\text{train}}, \tag{2}$$

where for $x \in \mathbb{R}^D$ and $z = \phi(x) \in \mathbb{R}^d$, the notation $P^e_\theta(X, Z)$ denotes a distribution over $\mathbb{R}^{D+d}$ such that $\Pr(X \in dx, Z \in dz) = P_X(x)\delta(z - \phi(x))dxdz$.

This formulation shares similarities with unsupervised representation learning approaches such as Variational Autoencoders (VAE) (Kingma & Welling, 2014) and Probabilistic PCA (Vidal et al., 2016), but crucially introduces an explicit constraint enforcing representation invariance across environments. Analogous to supervised IRM, the maximum likelihood term plays the role of the empirical risk, while the constraint enforces an invariance property, here defined as equality of feature distributions across environments.

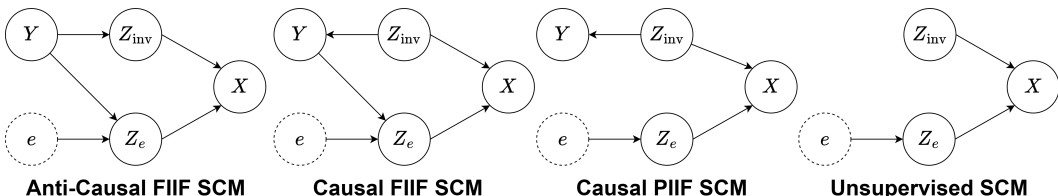

Figure 1: IRM Generative Structural Causal Models for supervised (left 3 figures) and unsupervised (right figure) cases

## 1.3 GENERATIVE CAUSAL GRAPHS

The foundational assumptions underlying this work, and much of the broader IRM literature, are rooted in causality theory (Peters et al., 2017; Schölkopf et al., 2021), specifically a particular family of causal generative processes. These processes are typically formalized using *Structural Causal Models* (SCMs), which characterize the cause-and-effect relationships among the variables in a system. In the IRM setting, prior works commonly assume SCMs that distinguish between "Fully Informative Invariant Features" (FIIF) and "Partially Informative Invariant Features" (PIIF) (Ahuja et al., 2021), as well as between causal and anti-causal label structures. A central concept across these variants is the decomposition of the latent space into two components: $Z_e$, the environment-dependent features, and $Z_{\text{inv}}$, the invariant features that remain stable across environments.

In this work, we introduce a new SCM tailored for the unsupervised setting, which we term *Unsupervised SCM*. This model generalizes previous assumptions, providing a unified framework that encompasses both FIIF and PIIF structures, as well as causal and anti-causal generative mechanisms. By doing so, it lays the foundation for unsupervised invariant representation learning under a broader and more flexible generative model.

## 2 RELATED WORK

The study of causality predates the formulation of Invariant Risk Minimization (IRM). In particular, Peters et al. (2015) established a connection between causal relationships and invariance principles, laying the foundation for the development of IRM. IRM was formally introduced by Arjovsky et al. (2019), along with its first approximating objective, IRMv1. Following this, substantial research has focused on designing improved objectives and algorithms. Notably, Zhou et al. (2022) and Lin et al. (2022) proposed methods that perform better in the over-parameterized regime, while Ahuja et al. (2021) introduced a bottleneck-based approach and also considered the differences between the FIIF and PIIF cases. Lin et al. (2022) also leveraged stochastic networks, similarly to our method, though their work remained within the supervised setting. In Salaudeen & Koyejo (2024), both the invariant component $Z_{\text{inv}}$ and the environment-dependent component $Z_e$ of the latent representation were parameterized and learned. We adopt a similar modeling choice, as data-point reconstruction and generation cannot be achieved without the environmental part of the latent space. A more detailed discussion on this observation appears in Section 4.3.

On the theoretical side, Rosenfeld et al. (2020) highlighted limitations of supervised IRM, demonstrating that under mild assumptions, an impractically large number of environments may be required to guarantee generalization to unseen environments. Wald et al. (2022) further showed that the interpolation property, common in over-parametrized learning algorithms, precludes invariance, providing theoretical justification for the strategies proposed in Lin et al. (2022) and Zhou et al. (2022). Finally, Toyota & Fukumizu (2023) established that, under suitable assumptions, the IRM objective indeed leads to environment-robust solutions.

Outside the IRM framework, Neria & Nir (2024) explore an unsupervised approach to learning representations that are optimized for downstream tasks rather than for robustness to distribution shifts. Unsupervised invariant representation learning was also studied prior to the introduction of IRM. For example, Lopez et al. (2018) and Moyer et al. (2018) also used the variational auto-

encoder framework toward this goal. Other notable works, relying on different sets of assumption and frameworks, include Sun et al. (2016) and Muandet et al. (2013).

# 3 PRINCIPAL INVARIANT COMPONENT ANALYSIS

The general objective for the unsupervised IRM problem, as given in equation equation 2, is to maximize the likelihood under the constraint of invariant featurization. Before tackling the general case, we focus on a simpler yet instructive setting: the linear and Gaussian case, where $X_e \sim P^e = \mathcal{N}(\mu_x^e, \Sigma_x^e) \ \forall e \in \mathcal{E}$. A well-known dimensionality reduction technique under similar assumptions is Principal Component Analysis (Vidal et al., 2016). PCA falls within the broader class of unsupervised learning algorithms, and in this section, we propose a variant tailored to the IRM setting, which we denote as *Principal Invariant Component Analysis* (PICA). PICA aims to eliminate "environmental dimensions," or equivalently, to find an invariant projection across environments.

Before we dive into the problem at hand, let us make an additional simplifying assumption. We assume that the data in each environment is mean-centered, i.e., $\mu_e = 0 \ \forall e \in \mathcal{E}$. This can be easily achieved by centering the data using $X_e \leftarrow X_e - E_{x \sim P^e}[X]$ for each environment. The resulting PICA optimization problem is

$$\max_u \sum_{e \in \mathcal{E}_{\text{train}}} E_{x \sim P^e}[(u^\top X)^2] \quad ; \quad \text{s.t.} \quad ||u||_2^2 = 1, \ P^i(u^\top X) = P^j(u^\top X) \ \forall i, j \in \mathcal{E}_{\text{train}}. \quad (3)$$

The objective seeks a vector $u$ such that the random variable's $u^\top X$ second moment (variance) is maximized across all training environments. Intuitively, this ensures that we retain the direction containing the most information across environments, similar to standard PCA. The constraint has two parts, first, $||u||_2^2 = 1$ eliminates scaling redundancy. The second constraint is where the IRM part of the problem comes into play, this constraint limits $u$ to be an invariant direction/dimension. Given the zero-mean assumption, the problem simplifies to

$$\max_u \sum_{e \in \mathcal{E}} u^\top \Sigma_x^e u \quad ; \quad \text{s.t.} \quad ||u||_2^2 = 1, \ u^\top \Sigma_x^i u = u^\top \Sigma_x^j u \ \forall i, j \in \mathcal{E}_{\text{train}}. \quad (4)$$

For simplicity, we focus on the two environments case $|\mathcal{E}_{\text{train}}| = 2$. In this case, the invariance constraint reduces to $u^\top \left( \Sigma_x^1 - \Sigma_x^2 \right) u = 0$, meaning that $u$ must lie in the null space of $(\Sigma_x^1 - \Sigma_x^2)$, i.e., $u \in \ker(\Sigma_x^1 - \Sigma_x^2)$. The objective, meanwhile, simplifies to $u^\top(\Sigma_x^1 + \Sigma_x^2)u$. Thus, the PICA solution can be found via the following two-step procedure:

1. Find $\mathcal{U} = \ker \left( \Sigma_x^1 - \Sigma_x^2 \right)$
2. Choose $u$ according to $\max_{u \in \mathcal{U}} u^\top \left( \Sigma_x^1 + \Sigma_x^2 \right) u$

The second step of the exact solution appears in the appendix. Finally, if we're interested in finding a dimensionality reduction scheme that keeps the $d_r \leq d_{\text{inv}}$ most varying invariant components, we can simply repeat step two for $d_r$ times, choosing each time the next best vector in the null space - the one which maximizes the objective.

---

**Algorithm 1** PICA (Principal Invariant Component Analysis)

---

1: **Input:** Covariance matrices $\Sigma_x^1, \Sigma_x^2$; desired reduced invariant dimension $d_r \leq d_{\text{inv}}$
2: **Output:** Matrix $U_r \in \mathbb{R}^{n \times d_r}$ containing the top $d_r$ invariant principal directions
3: Initialize $U_r \leftarrow \emptyset$
4: Compute the invariant subspace: $\mathcal{U} = \{u \in \mathbb{R}^n | (\Sigma_x^1 - \Sigma_x^2)u = 0\}$
5: **for** $i = 1$ to $d_r$ **do**
6:     Find $u^* = \arg\max_{u \in \mathcal{U}} u^\top (\Sigma_x^1 + \Sigma_x^2)u$
7:     Update $U_r \leftarrow \text{concat}(U_r, u^*)$
8:     Update $\mathcal{U} \leftarrow \mathcal{U} \setminus \{u^*\}$
9: **end for**
10: **Return** $U_r$

---

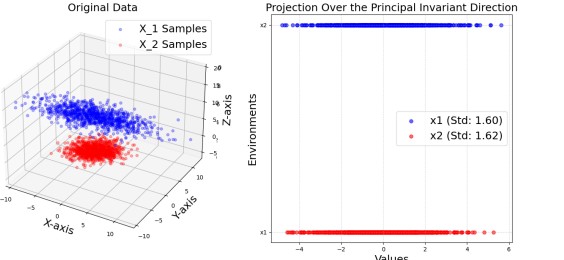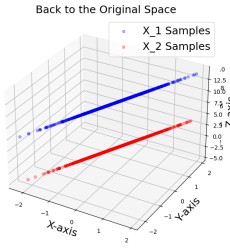

Figure 2: Output of the PICA algorithm with $d_r = 1$ on the synthetic dataset. The projection captures the invariant component shared across both environments.

## 3.1 EXPERIMENTING WITH PICA

We illustrate PICA on a simple synthetic example based on the generative process

$$X_e = \mu_e(e) + A_{\text{inv}}Z_{\text{inv}} + A_e Z_e + \epsilon,$$

where

$$\mu_e(1) = [0,0,0]^\top, \ \mu_e(2) = [0,0,5]^\top, \ A_{\text{inv}} = [1,1,1]^\top, \ A_e = [1,1,-1]^\top,$$
$$\sigma_e^2(1) = 10, \ \sigma_e^2(2) = 2, \ Z_{\text{inv}} \sim \mathcal{N}(0,1), \ Z_e \sim \mathcal{N}(0,\sigma_e^2(e)), \ \epsilon \sim \mathcal{N}(0,0.025).$$

We generate 1000 samples of $X_1$ and 1000 samples of $X_2$. We then apply PICA with target reduced dimension $d_r = 1$. The results are shown in Figure 2. Although the original data is divided between two environments that are clearly characterized by different covariance matrices, the projection of the data exhibits a constant distribution across environments.

## 4 VARIATIONAL INVARIANT AUTO-ENCODER

The main algorithm introduced in this work is the *Variational Invariant Autoencoder* (VIAE). It follows the standard VAE (Kingma & Welling, 2014) framework, but with a key modification- it is specifically designed to recover the structure of the proposed "unsupervised" SCM shown in Figure 1(right). The block diagram in Figure 3 reflects this structure. The algorithm's architecture has the following favorable properties:

1. **Factorized Latent Space:** VIAE explicitly factorizes the latent space into two subspaces: an invariant component $Z_{\text{inv}}$ and an environment-specific component $Z_e$.

2. **Latent Space Interventions:** VIAE enables interventions on the latent space by sampling $Z_e$ from different priors.

3. **Bottleneck Architecture:** VIAE enforces a narrow latent representation, acting as an information bottleneck. This aligns with findings in IRM literature (e.g., (Ahuja et al., 2021)) suggesting that bottlenecks improve the identification of invariant predictors.

Where properties 2 and 3 are inherited from the baseline VAE framework. VIAE contains one decoder, one Invariant encoder to produce the Invariant part of the latent space, and $|\mathcal{E}_{\text{train}}|$ environmental encoders (one for each environment), to produce the environmental part of the latent space. Another way to look at it is that the decoder's and invariant encoder's parameters are shared across environments, while the environmental encoder's parameters are unique for each environment.

By utilizing the causality constraints induced by the unsupervised SCM, we define the latent space and the encoder and decoder probabilities.

For the **Encoder**, for a given environment $e$, the **prior** over latent variables is

$$P^e(Z_{\text{inv}}, Z_e) = P^e(Z_{\text{inv}})P^e(Z_e) = P(Z_{\text{inv}})P^e(Z_e),$$

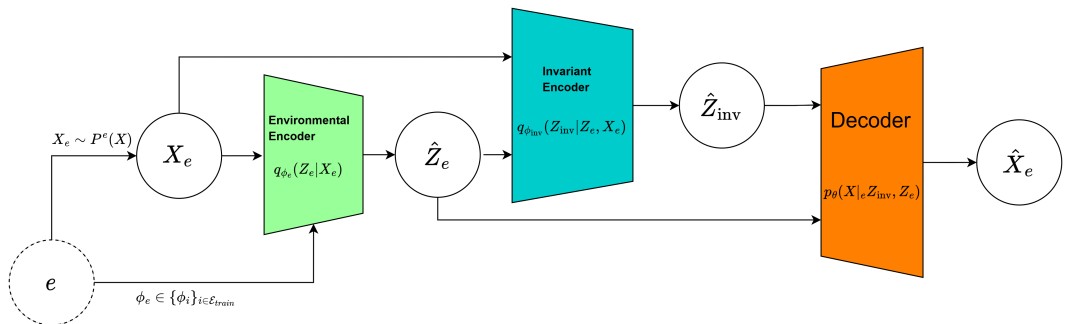

Figure 3: VIAE architecture. A shared invariant encoder produces $Z_{\text{inv}}$, while environment-specific encoders produce $Z_e$. The decoder reconstructs $X$ from both components.

where we used the causal graph to determine that $Z_{\text{inv}} \perp\!\!\!\perp Z_e, \ Z_{\text{inv}} \perp\!\!\!\perp e$. For the invariant encoder, we use the standard prior $Z_{\text{inv}} \sim \mathcal{N}(0, I)$. As for each environmental encoder, we introduce an environment-specific prior $Z_e \sim \mathcal{N}(\mu_e(e), I), \quad \mu_e(i) \perp \mu_e(j) \ \forall i, j \in \mathcal{E}_{\text{all}}$, enabling controlled sampling from a chosen environment during generation. The encoder **posterior** probability factorizes as

$$P^e(Z_{\text{inv}}, Z_e|X) = P^e(Z_{\text{inv}}|Z_e, X)P^e(Z_e|X) = P(Z_{\text{inv}}|Z_e, X)P^e(Z_e|X).$$

This factorization reflects two key causal properties (Peters et al., 2017). *(i)* Given $Z_e$, $Z_{\text{inv}}$ is conditionally independent of the environment $e$. *(ii)* $Z_{\text{inv}}$ and $Z_e$ become statistically dependent when conditioning on $X$, due to the collider structure $Z_{\text{inv}} \to X \leftarrow Z_e$.

As a consequence, the invariant encoder takes both $X$ and $Z_e$ as input, while each environmental encoder depends only on $X$.

The relevant distribution for the **Decoder** is the **conditional likelihood** probability $P^e(X|Z_{\text{inv}}, Z_e)$. Invoking the causal structure, we note that when conditioned on $Z_{\text{inv}}$ and $Z_e$, the environment $e$ provides no additional information about $X$. Thus

$$P^e(X|Z_{\text{inv}}, Z_e) = P(X|Z_{\text{inv}}, Z_e) = P(X|Z).$$

This is a **causal mechanism**, meaning that it's independent of its source $Z$ distribution, which means that any interventions (changes in the distribution) of $Z$ won't affect the decoder. Note that the decoder does not receive any explicit information of the environment, as the invariant (causal) mechanism is stable across different environments, as assumed in causality theory.

## 4.1 DATASETS

We evaluate VIAE on two synthetic datasets of our own design, inspired by benchmarks commonly used in the IRM and domain adaptation literature (Gulrajani & Lopez-Paz, 2020).

**SMNIST:** We introduce SMNIST (Squares MNIST), a variant of MNIST (Bottou et al., 1994) with artificial spurious features. It includes two training environments, each using half of the MNIST training set. In $e = 1$, a 7×7 white square is added to the top-left corner; in $e = 2$, the square appears in the bottom-right. Two analogous test environments use the MNIST test set with squares in the top-right ($e = 3$) and bottom-left ($e = 4$) corners, respectively.

**SCMNIST:** Inspired by the commonly known CMNIST ((Arjovsky et al., 2019), (Gulrajani & Lopez-Paz, 2020) and many more), we define SCMNIST (Single Colored MNIST). SCMNIST contains two training environments and one test environment. In training, MNIST digits are encoded into either the red RGB channel for $e = 1$ or to the green RGB channel for $e = 2$ (others set to zero). The test environment ($e = 3$) encodes digits in the blue channel.

For experiments involving the more realistic **CelebA** dataset, please see The fariness section D.

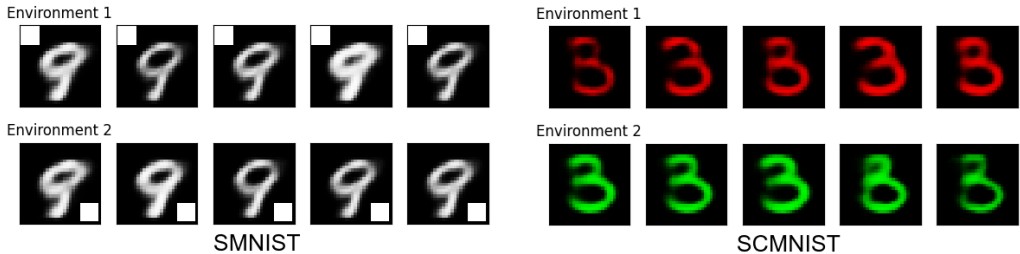

Figure 4: Generated samples conditioned on a fixed $Z_{\text{inv}}$. Top row: samples with different $Z_e$ drawn from $P^1(Z_e)$. Bottom row: samples with different $Z_e$ sampled from $P^2(Z_e)$. Left side: SMNIST dataset, right side: SCMNIST dataset. Invariant features (in our case, the digits) are preserved for all samples, with stable environment for each row.

## 4.2 SAMPLE GENERATION

To illustrate the capabilities of VIAE, we demonstrate that for a single instance of $Z_{\text{inv}}$, sampled from its prior, the output generated samples from the decoder have the same invariant properties regardless of $Z_e$. On the other hand, the generated samples environment is consistent with the chosen prior of $Z_e$, which means that the decoder is able to correctly produce samples from a specific environment without it being provided any explicit information over the desired environment. This is demonstrated in Figure 4.

## 4.3 ENVIRONMENT TRANSFER

Beyond sample generation, we leverage the VIAE framework to address the IRM problem. Considering our unsupervised framework, and specifically the VAE settings, some modifications to the IRM goals are necessary. Traditionally, IRM seeks an invariant representation $\phi(X)$ such that a downstream predictor $w \circ \phi(X)$ performs well across all environments. In our case, the invariant encoder naturally plays the role of the featurizer such that $\hat{Z}_{\text{inv}} = \phi(X)$, but the decoder cannot be interpreted as a classifier nor a regressor. This raises an important question - what should a satisfying restoration look like? and/or what should the IRM objective be in our setting? We propose that the IRM problem can be considered as "solved" if all data-points can be "transferred" to a single environment, while preserving their invariant content. Since distribution shifts are assumed to arise only across environments, if we manage to take a dataset that is divided between environments, and convert it to an equivalent dataset that contains a single environment, then the problem reduces to an ordinary learning problem, and the IRM problem can be considered as solved. More formally, the goal is to transform each data-point from its original environment $e_s \in \mathcal{E}_{\text{all}}$ into an equivalent sample in a predetermined target environment $e_t \in \mathcal{E}_{\text{train}}$, while maintaining the invariant features $Z_{\text{inv}}$ and adapting only the environmental ones $Z_e$. That is, we would like to generate $\hat{X}_{e_t} \sim P^{e_t}(X|Z_{\text{inv}} = \text{ENC}_{\text{inv}}(X_{e_s}, Z_{e_s}))$, where $\text{ENC}_{\text{inv}}(\cdot)$ is the invariant encoder function. To illustrate, consider the well-known "camels and cows" example, where cows often appear in green pastures and camels in deserts (as appears in many works, including Arjovsky et al. (2019)). Such dataset contains spurious correlations between the classes and the image background. However, if we can transform all images so that both cows and camels appear in the same environment (e.g., desert), these spurious background cues lose their predictive power. In such a scenario, we argue that the IRM objective can be considered as solved, not by removing spurious features entirely, but by aligning them across environments. An alternative, seemingly more "straightforward" idea might be to generate "purely invariant" samples. However, we argue that such representations are ill-defined in many domains. For instance, in medical imaging, brightness might be a spurious factor correlated with equipment rather than pathology. What does a brightness-invariant X-ray look like? Removing brightness altogether would destroy the data. Thus, transferring all data to a single environment, rather than stripping away the environmental features, serves as a practical alternative that gets the job done. This task resembles domain adaptation as explored in Tzeng et al. (2017) and Zhu et al. (2017), but with a key distinction - VIAE enables, in some circumstances, to perform environment

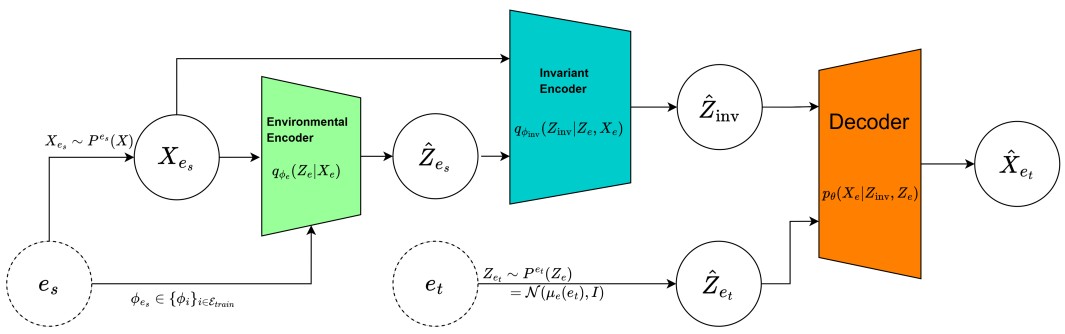

Figure 5: Environment Transfer for $e_s \in \mathcal{E}_{\text{train}}$

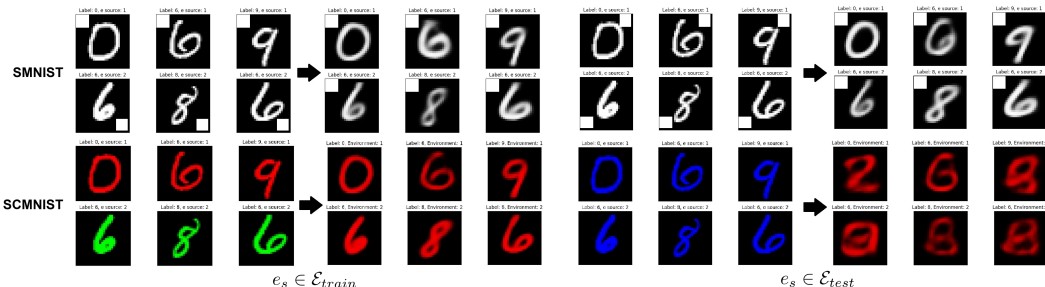

Figure 6: Environment transfer examples. $e_s \in \mathcal{E}_{\text{train}}$ for the left part and $e_s \in \mathcal{E}_{\text{test}}$ for the right part, demonstrated for the SMNIST (up) and SCMNIST (down) datasets.

transfer from unseen source environments $e_s \in \mathcal{E}_{\text{test}}$. To our knowledge, previous methods typically require both source and target environments to be seen during training.

### 4.3.1 ENVIRONMENT TRANSFER FOR SEEN & UNSEEN ENVIRONMENTS

Consider the case in which the **source environment has been seen during training**, i.e., $e_s \in \mathcal{E}_{\text{train}}$. Under this assumption, environment transfer can be performed using the following procedure

1. Sample a data-point from the source environment $X_{e_s} \sim P^{e_s}$, $e_s \in \mathcal{E}_{\text{train}}$
2. Pass $X_{e_s}$ through the environmental encoder corresponding to the source environment to obtain the source environmental features $\hat{Z}_{e_s} \sim P^{e_s}(Z_e | X_{e_s})$
3. Use $X_{e_s}$ and $\hat{Z}_{e_s}$ as inputs to the invariant encoder to get $\hat{Z}_{\text{inv}}$ : $\hat{Z}_{\text{inv}} \sim P(Z_{\text{inv}} | X_{e_s}, \hat{Z}_{e_s})$
4. Sample $\hat{Z}_{e_t}$ from the target environment encoder prior: $\hat{Z}_{e_t} \sim P^{e_t}(Z_e) = \mathcal{N}(\mu_e(e_t), I)$
5. Feed $\hat{Z}_{\text{inv}}$ and $\hat{Z}_{e_t}$ to the Decoder to get $\hat{X}_{e_t}$: $\hat{X}_{e_t} = \text{Dec}(\hat{Z}_{\text{inv}}, \hat{Z}_{e_t})$

This process is illustrated in Figure 5 and demonstrated in Figure 6(left).

When the source environment is **not part of the training set**, i.e., $e_s \in \mathcal{E}_{\text{test}}$, we encounter a fundamental limitation - we do not possess an environmental encoder trained for $e_s$. Consequently, generalization to arbitrary unseen environments is not guaranteed in the general case. However, for some easier scenarios, we may attempt to estimate the environmental features by leveraging the existing environmental encoders. Specifically, by using the approximation

$$\hat{Z}_{e_s} = \frac{1}{|\mathcal{E}_{\text{train}}|} \sum_{e \in \mathcal{E}_{\text{train}}} Z_e \quad ; \quad Z_e \sim P^e(\cdot | X_{e_s}).$$

That is, we pass $X_{e_s}$ through every one of the environmental encoders and average their outputs to get an estimation of $Z_{e_s}$. As shown in Figure 6 (right part), this approach works reasonably

Table 1: Accuracy of the four defined classifiers, on test data-points from training environments. Reported results are mean $\pm$ standard deviation over 10 runs.

| Classifier | SMNIST | SCMNIST |
|---|---|---|
| $\hat{Y}_{I2L}$ | $0.845 \pm 0.050$ | $0.832 \pm 0.072$ |
| $\hat{Y}_{e2L}$ | $0.362 \pm 0.041$ | $0.345 \pm 0.045$ |
| $\hat{e}_{I2e}$ | $0.556 \pm 0.066$ | $0.583 \pm 0.055$ |
| $\hat{e}_{e2e}$ | $1.0 \pm 0$ | $1.0 \pm 0$ |

well for the easier case of the SMNIST dataset. However, it fails for the more complex SCMNIST dataset. This disparity can be understood using the insights from Rosenfeld et al. (2020). In essence, generalization to unseen environments is possible only when the training environments sufficiently "cover" the space of all possible environments. When this coverage is lacking, such generalization is fundamentally out of reach. For example, in the SCMNIST case, for all of the training environments, the blue color channel always equals zero. This means that even in the basic linear - algebraic sense, the training environments don't span the "blue dimension".

## 4.4 BACK TO SUPERVISED LEARNING

Recall the original supervised IRM problem as formalized in equation 1. The goal is to learn an invariant features extractor $\hat{Z}_{\text{inv}} = \phi(X_e)$ such that the prediction $\hat{Y} = w \circ \phi(X_e)$ is invariant. Although our own approach does not adopt this supervised setup, there is a natural similarity between the invariant feature extractor in supervised IRM and the encoders in our model. The question we would like to answer is whether our encoders perform well as such feature extractors, and if so, does that mean our completely unsupervised algorithm solves the original supervised IRM problem? To investigate this, consider the feature extractor $\hat{Z} = \left[ \hat{Z}_{\text{inv}}, \ \hat{Z}_e \right]^{\top} = \phi(X_e)$. Where $\hat{Z}_e$ is the output of the environmental encoder and $\hat{Z}_{\text{inv}}$ the output of the invariant encoder. We train four linear classifiers on top of these features:

$$\hat{Y}_{I2L} = w_{I2L}^{\top} Z_{\text{inv}} \qquad \text{(Label prediction from invariant features)}$$
$$\hat{Y}_{e2L} = w_{e2L}^{\top} Z_e \qquad \text{(Label prediction from environmental features)}$$
$$\hat{e}_{I2e} = w_{I2e}^{\top} Z_{\text{inv}} \qquad \text{(Environment prediction from invariant features)}$$
$$\hat{e}_{e2e} = w_{e2e}^{\top} Z_e \qquad \text{(Environment prediction from environmental features)}$$

By "environment prediction" we mean that the original environment from which each data point came serves as a label, and the objective is to perform classification based on it. This experiment aims to examine whether the model successfully learns a separated latent space, with invariant information captured in $\hat{Z}_{\text{inv}}$ and environmental information in $\hat{Z}_e$. We evaluate the performance of the different classifiers on test data from the training environments $\mathcal{E}_{\text{train}}$. We train the VIAE network from scratch 10 times. After each training run, we fit the aforementioned linear classifiers on top of the encoders output and evaluate their accuracy on the test set. The results are reported in Table 1. Let's analyze these results. The label classifier using invariant features ($\hat{Y}_{I2L}$) achieves high accuracy (around $0.84 - 0.83$), indicating that the invariant encoder successfully retains the relevant class information, in our case, the digit identity, which is invariant by construction. In contrast, the label classifier using environmental features ($\hat{Y}_{e2L}$) achieves much lower accuracy ($0.36 - 0.34$). While this is higher than random chance ($0.1$), suggesting some residual label correlation in the environmental features, it is significantly worse than the invariant-based classifier. The third classifier ($\hat{e}_{I2e}$) attempts to predict the environment from the invariant features. It yields near-random accuracy ($0.55 - 0.58$), close to the baseline of $0.5$ for a trivial classifier with no information. We argue that this is the most important result in this section. Much of the IRM literature focuses on filtering out spurious (environmental) features to achieve robust generalization - even under distribution shifts where spurious correlations invert (e.g., cows on sandy beaches at test time after only seeing cows on grass and camels in deserts during training). While this result does not theoretically guarantee the absence of spurious correlations in $\hat{Z}_{\text{inv}}$, it provides a strong empirical indication that the invariant space is indeed, invariant. Finally, the fourth classifier ($\hat{e}_{e2e}$) predicts the environment from

environmental features and achieves perfect accuracy (1.0) for both datasets. This demonstrates that the environmental latent space captures all the information needed to distinguish between environments, exactly as intended. These results further demonstrate that our model achieves the desired separation in latent space, and motivates the possibility of using similar methods for downstream supervised IRM objectives.

## 5 Conclusions and Future Work

In this paper, we introduced an unsupervised approach to the IRM problem. We proposed two algorithms, each designed to solve the problem under different assumptions and requirements. Looking ahead, we hope this new unsupervised framework opens the door to novel insights and solutions for IRM, ones that are difficult or impossible to achieve within the well-studied supervised paradigm. We outline two concrete directions for future research. *(i)* Develop a theoretically complete scheme for performing environment transfer from previously unseen environments, i.e., $e_s \in \mathcal{E}_{\text{test}}$. *(ii)* Incorporate more advanced learning architectures to improve empirical performance. Regarding the first direction, we believe that meta-learning approaches (such as MAML (Finn et al., 2017)) may enable effective unseen-environment transfer in few-shot or one-shot settings. For instance, it may be possible to fine-tune environmental encoders to adapt to a new environment using only a few examples. For true zero-shot transfer, however, a different model architecture is probably necessary. As for the second direction, our work relied on the vanilla VAE architecture as baseline for the VIAE scheme. Leveraging more modern generative models (such as GANS (Goodfellow et al., 2014) and diffusion models (Ho et al., 2020)) could allow us to extend the success achieved on SMNIST, SCMNIST and CelebA to more complex and realistic datasets.

The code we used for the experiments is available at https://github.com/Yotamnor/UIRM.

## 6 Acknowledgments

This research was partially supported by grant no. 2022330 from the United States - Israel Binational Science Foundation (BSF), by Israel Science Foundation (ISF) grant no. 1693/22, and by the Skillman chair (RM).

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

# Appendix

## A PRINCIPAL INVARIANT COMPONENT ANALYSIS

### A.1 ALGORITHM'S SOLUTION

In this section, we'll focus on solving the two-steps procedure of the PICA algorithm described in 3. The first step is done by simply solving for $\left(\Sigma_x^1 - \Sigma_x^2\right) u = 0$. The solution to the second step can be obtained by setting $u = Uv$, where $v \in \mathbb{R}^{d_{\mathrm{inv}}}$ is a normalized vector, and $U \in \mathbb{R}^{D \times d_{\mathrm{inv}}}$ is a matrix whose columns span $\mathcal{U}$. Substituting this term into the optimization objective yields

$$\max_{v} \ v^\top U^\top \left(\Sigma_x^1 + \Sigma_x^2\right) Uv$$
$$\text{s.t.} \quad ||v||_2^2 = 1.$$

This is equivalent to the standard PCA objective, with $U^\top \left(\Sigma_x^1 + \Sigma_x^2\right) U$ playing the role of the covariance matrix. The optimal $u$ is given by $u = Uv$, where $v$ is the eigenvector associated with the largest eigenvalue of $U^\top (\Sigma_x^1 + \Sigma_x^2) U$. For the subsequent $u_2, u_3, \ldots, u_{d_r}$ dimensions, we simply set $u_i = Uv_i$, where $v_i$ is the $i$-th eigenvector of $U^\top (\Sigma_x^1 + \Sigma_x^2) U$, corresponding to the $i$-th largest eigenvalue.

As a final note, we address the case with more than two environments, $|\mathcal{E}| > 2$. Although the derivations for this case were not carried out as part of this work, we believe that they can be conducted as a natural extension for the two environments setting. The solution in this case would most likely involve finding the direction that maximizes the quadratic form over the sum of all environments, subject to the constraint that the projection lies in the intersection of the null spaces of the pairwise difference matrices between all environments. The resulting two-step procedure is as follows:

1. Compute

$$\mathcal{U} = \bigcap_{i,j \in \mathcal{E}_{\mathrm{train}}} \ker\left(\Sigma_x^i - \Sigma_x^j\right)$$

2. Choose $u$ according to

$$\max_{u \in \mathcal{U}} u^\top \left(\sum_{e \in \mathcal{E}_{\mathrm{train}}} \Sigma_x^e\right) u$$

A complete derivation and empirical validation of this proposed extension are left for future work.

### A.2 ANALYTICAL DERIVATION OF PICA

We motivate the PICA algorithm using the following assumed underlying linear Gaussian model

$$X_e = \mu_e(e) + A_{\mathrm{inv}} Z_{\mathrm{inv}} + A_e Z_e + \epsilon \tag{5}$$

Where

$$X_e \in \mathbb{R}^D$$
$$Z_{\mathrm{inv}} \in \mathbb{R}^{d_{\mathrm{inv}}}, \quad Z_{\mathrm{inv}} \sim \mathcal{N}(0, I)$$
$$Z_e \in \mathbb{R}^{d_e}, \quad Z_e \sim \mathcal{N}(0, I\sigma_e^2(e))$$
$$\epsilon \sim \mathcal{N}(0, I\sigma_\epsilon^2) \quad \text{Gaussain noise}$$
$$\text{From the causality graph 1: } Z_{\mathrm{inv}} \perp\!\!\!\perp Z_e, \ A_i = A_j \ \forall i, j \in \mathcal{E}$$

Here, $\mu_e(e)$, $A_{\mathrm{inv}}$, $A_e$ are deterministic with appropriate dimensions. Importantly, $\mu_e(e)$ and $\sigma_e^2(e)$ vary across environments, hence are written explicitly as functions of $e$, while the matrices $A_{\mathrm{inv}}$ and

$A_e$ remain stable across environments. From this model, the covariance in environment $e$ can be derived as

$$\Sigma_x^e = A_{\text{inv}} A_{\text{inv}}^\top + \sigma_e^2(e) A_e A_e^\top + \sigma_\epsilon^2 I$$

Consequently, we have

$$\Sigma_x^1 - \Sigma_x^2 = (\sigma_e^2(1) - \sigma_e^2(2)) A_e A_e^\top, \quad \Sigma_x^1 + \Sigma_x^2 = 2 A_{\text{inv}} A_{\text{inv}}^\top + (\sigma_e^2(1) + \sigma_e^2(2)) A_e A_e^\top + 2\sigma_\epsilon^2 I$$

First, observe that when $\sigma_e^2(1) = \sigma_e^2(2)$ the problem reduces to a single environment case and we simply perform standard PCA as demonstrated in Vidal et al. (2016). Thus, we henceforth assume $\sigma_e^2(1) \neq \sigma_e^2(2)$. We can see that $\Sigma_x^1 - \Sigma_x^2$ contains only the environmental data component. Therefore, by choosing principal components from its null space, we remove environment-dependent dimensions. Further evaluating the objective yields

$$u^\top \left( \Sigma_x^1 + \Sigma_x^2 \right) u = 2u^\top \left( A_{\text{inv}} A_{\text{inv}}^\top + \sigma_\epsilon^2 I \right) u + (\sigma_e^2(1) + \sigma_e^2(2)) u^\top \left( A_e A_e^\top \right) u$$
$$= 2u^\top \left( A_{\text{inv}} A_{\text{inv}}^\top + \sigma_\epsilon^2 I \right) u.$$

Where the second equality is due to the second term in vanishing for $u$ in the null space of $A_e A_e^\top$. Assuming that the invariant information signal $A_{\text{inv}} Z_{\text{inv}}$ dominates the noise $\epsilon$, we get that maximizing the objective under the IRM constraint will yield a vector in the direction that maximizes the invariant variance, as we would expect from an IRM version of PCA.

### A.3 PROBABILISTIC PICA ALGORITHM

Building on the derivation presented in Vidal et al. (2016), we proceed to develop a generative probabilistic (linear) model tailored to our IRM setting. By learning a generative model, we can sample new synthetic examples from the underlying data distribution.

Consider the generative model presented in 5. The environment-specific expectation and covariance matrix are

$$\mu_{X_e} = \mu_e(e), \quad \Sigma_x^e = A_{\text{inv}} \Sigma_{\text{inv}} A_{\text{inv}}^\top + A_e \Sigma_e(e) A_e^\top + \Sigma_\epsilon = A_{\text{inv}} A_{\text{inv}}^\top + \sigma_e^2(e) A_e A_e^\top + \sigma_\epsilon^2 I.$$

For simplicity, assume from now on that $d_{\text{inv}} = d_e = d$.

In order to reconstruct the generative model, the objective in *Probabilistic PICA* is to extract the model's parameters $\mu_e(e)$, $\sigma_e^2(e)$, $A_{\text{inv}}$, $A_e$, $\sigma_\epsilon^2$ from the population mean and the population covariance. An immediate result is that for every environment, the environmental mean simply equals to the population mean over the environment. Consider the covariance matrix, Since both $A_{\text{inv}} A_{\text{inv}}^\top$ and $A_e A_e^\top$ have a rank of $d$, their $D - d$ smallest eigenvalues are equal to zero, which means that the $D - 2d$ smallest eigenvalues of the covariance matrix are equal to $\sigma_\epsilon^2$.

In order to simplify the upcoming derivations, we consider from now on the two environments case $|\mathcal{E}| = 2$.

Consider

$$\Sigma_x^1 - \Sigma_x^2 = (\sigma_e^2(1) - \sigma_e^2(2)) A_e A_e^\top$$

Without loss of generality, assume that $\sigma_e^2(1) \geq \sigma_e^2(2)$, which yields that the subtraction matrix $\Sigma_x^1 - \Sigma_x^2$ is positive semi-definite (PSD). The special case of $\sigma_e^2(1) = \sigma_e^2(2)$ corresponds to the degenerate case in which there is no distribution shift. Therefore, we'll assume $\sigma_e^2(1) > \sigma_e^2(2)$. By using the spectral decomposition of the subtraction matrix, we get that the estimation for $A_e$ is

$$\hat{A}_e = \frac{1}{\sqrt{\sigma_e^2(1) - \sigma_e^2(2)}} U_s \Lambda_s^{1/2}.$$

With $U_s$ containing the $d$ eigen vectors corresponding the the $d$ highest eigenvalues of $\Sigma_x^1 - \Sigma_x^2$ and $\Lambda_s$ is a diagonal matrix containing the eigenvalues of $\Sigma_x^1 - \Sigma_x^2$. Consider the mean of the covariance matrices

$$\frac{\Sigma_x^1 + \Sigma_x^2}{2} = A_{\text{inv}} A_{\text{inv}}^\top + \frac{\sigma_e^2(1) + \sigma_e^2(2)}{2} A_e A_e^\top + \sigma_\epsilon^2 I$$

$$A_{\text{inv}} A_{\text{inv}}^\top = \frac{\Sigma_x^1 + \Sigma_x^2}{2} - \frac{\sigma_e^2(1) + \sigma_e^2(2)}{2(\sigma_e^2(1) - \sigma_e^2(2))} (\Sigma_x^1 - \Sigma_x^2) - \sigma_\epsilon^2 I$$

Define

$$\mathcal{M}(\sigma_e^2(1), \sigma_e^2(2)) \triangleq A_{\text{inv}} A_{\text{inv}}^\top = \frac{\Sigma_x^1 + \Sigma_x^2}{2} - \frac{\sigma_e^2(1) + \sigma_e^2(2)}{2(\sigma_e^2(1) - \sigma_e^2(2))} (\Sigma_x^1 - \Sigma_x^2) - \sigma_\epsilon^2 I$$

The resulting estimation of $A_{\text{inv}}$ is

$$\hat{A}_{\text{inv}} = U_a \Lambda_a^{1/2}.$$

With $U_a$ containing the $d$ eigen vectors corresponding the the $d$ highest eigenvalues of $\mathcal{M}\left(\sigma_e^2(1), \sigma_e^2(2)\right)$ and $\Lambda_a$ is a diagonal matrix containing the eigenvalues of $\mathcal{M}\left(\sigma_e^2(1), \sigma_e^2(2)\right)$ arranged in decreasing order. In order to complete the generative model, an estimation for $\sigma_e^2(1)$ and $\sigma_e^2(2)$ is needed. Without loss of generality, we can set $\hat{\sigma}_e^2(1) = 1$, as this is simply a matter of rescaling $\hat{A}_e$ appropriately. Now, all that is left is to find a population-level term for $\sigma_e(2)$. Consider the following term:

$$
\begin{aligned}
\frac{\text{tr}(\Sigma_x^1 - \Sigma_x^2)}{\text{tr}(\Sigma_x^1)} &= \frac{(\sigma_e^2(1) - \sigma_e^2(2))\text{tr}(A_e A_e^\top)}{\text{tr}(A_{\text{inv}} A_{\text{inv}}^\top) + \sigma_e^2(1)\text{tr}(A_e A_e^\top) + \sigma_\epsilon^2 D} \\
&= \frac{(1 - \sigma_e^2(2))\text{tr}(A_e A_e^\top)}{\text{tr}(A_{\text{inv}} A_{\text{inv}}^\top) + \text{tr}(A_e A_e^\top) + \sigma_\epsilon^2 D}.
\end{aligned}
\tag{6}
$$

In order to proceed, we must make further assumptions. Consider the case for which the environmental signal is significantly stronger than the invariant one, this could also be looked at as assuming that most of the information originates from the environmental features. This assumption reflects a more challenging IRM setup, where a weaker invariant signal is buried in much stronger environmental "noise". Consequently, assume that the environmental features are "stronger" in the following sense:

$$\text{tr}(A_e A_e^\top) \gg \text{tr}(A_{\text{inv}} A_{\text{inv}}^\top) \gg \sigma_\epsilon^2 D.$$

Using the above assumption together with $\sigma_e^2(1) = 1$, we get from 6 that

$$\hat{\sigma}_e^2(2) \approx 1 - \frac{\text{tr}(\Sigma_x^1 - \Sigma_x^2)}{\text{tr}(\Sigma_x^1)} = \frac{\text{tr}(\Sigma_x^2)}{\text{tr}(\Sigma_x^1)}.$$

Further evidence that this value represents a form of "worst-case" scenario can be obtained by noting that both $\text{tr}(A_{\text{inv}} A_{\text{inv}}^\top)$ and $\sigma_\epsilon^2 D$ are positive, meaning that

$$1 = \sigma_e^2(1) \geq \sigma_e^2(2) \geq \frac{\text{tr}(\Sigma_x^2)}{\text{tr}(\Sigma_x^1)}.$$

Consider the assumption $\sigma_e^2(2) < \sigma_e^2(1)$. By setting $\sigma_e^2(2)$ to its lower bound, the resulting model corresponds to the largest possible distribution shift between environments for the given data.

## B  VARIATIONAL INVARIANT AUTO-ENCODER

### B.1  ELBO DEVELOPMENT

First introduced by Kingma & Welling (2014), the Evidence Lower Bound (ELBO) in our setting is given by

$$E_{q_\phi(Z|X,e)}[\log p_\theta(X|Z)] - \text{KL}[q_\phi(Z|X,e) \parallel p_\theta(Z|e)]$$

For notational convenience in the derivations of this section, we treat the environment as a given conditional variable, writing $p_\theta(\cdot \mid e)$ rather than embedding it in the probability function as in $p_\theta^e(\cdot)$.

As a reminder from the encoder derivation in section 4

$$q_\phi(Z|X,e) = q_\phi(Z_{\text{inv}}, Z_e|X,e) = q_\phi(Z_e|X,e)q_\phi(Z_{\text{inv}}|Z_e, X),$$

we get

$$
\begin{aligned}
&E_{q_\phi(Z|X,e)}[\log p_\theta(X|Z)] - \text{KL}[q_\phi(Z|X,e) \parallel p_\theta(Z|e)] \\
=&E_{q_\phi(Z_{\text{inv}}, Z_e|X,e)}[\log p_\theta(X|Z_{\text{inv}}, Z_e)] - \text{KL}[q_\phi(Z_e|X,e)q_\phi(Z_{\text{inv}}|Z_e, X) \parallel p_\theta(Z_e|e)p_\theta(Z_{\text{inv}})]
\end{aligned}
$$

Consider on the KL term

$$\mathrm{KL}[q_\phi(Z_e|X,e)q_\phi(Z_{\mathrm{inv}}|Z_e,X) \parallel p_\theta(Z_e|e)p_\theta(Z_{\mathrm{inv}})]$$

$$= \int_{z_e} \int_{z_{\mathrm{inv}}} q_\phi(Z_{\mathrm{inv}}|Z_e,X)q_\phi(Z_e|X,e) \log \left[ \frac{q_\phi(Z_{\mathrm{inv}}|Z_e,X)q_\phi(Z_e|X,e)}{p_\theta(Z_e|e)p_\theta(Z_{\mathrm{inv}})} \right] dz_{\mathrm{inv}} dz_e$$

$$= \int_{z_e} \int_{z_{\mathrm{inv}}} q_\phi(Z_{\mathrm{inv}}|Z_e,X)q_\phi(Z_e|X,e) \log \left[ \frac{q_\phi(Z_e|X,e)}{p_\theta(Z_e|e)} \right] dz_{\mathrm{inv}} dz_e$$

$$+ \int_{z_e} \int_{z_{\mathrm{inv}}} q_\phi(Z_{\mathrm{inv}}|Z_e,X)q_\phi(Z_e|X,e) \log \left[ \frac{q_\phi(Z_{\mathrm{inv}}|Z_e,X)}{p_\theta(Z_{\mathrm{inv}})} \right] dz_e dz_{\mathrm{inv}}$$

$$= \int_{z_e} q_\phi(Z_e|X,e) \log \left[ \frac{q_\phi(Z_e|X,e)}{p_\theta(Z_e|e)} \right] \int_{z_{\mathrm{inv}}} q_\phi(Z_{\mathrm{inv}}|Z_e,X) dz_{\mathrm{inv}} dz_e$$

$$+ \int_{z_e} q_\phi(Z_e|X,e) \int_{z_{\mathrm{inv}}} q_\phi(Z_{\mathrm{inv}}|Z_e,X) \log \left[ \frac{q_\phi(Z_{\mathrm{inv}}|Z_e,X)}{p_\theta(Z_{\mathrm{inv}})} \right] dz_{\mathrm{inv}} dz_e$$

$$= \mathrm{KL}(q_\phi(Z_e|X,e) \parallel p_\theta(Z_e|e)) + E_{q_\phi(Z_e|X,e)}[\mathrm{KL}(q_\phi(Z_{\mathrm{inv}}|Z_e,X) \parallel p_\theta(Z_{\mathrm{inv}}))]$$

The total ELBO term is

$$E_{q_\phi(Z|X,e)}[\log p_\theta(X|Z)] - \mathrm{KL}(q_\phi(Z_e|X,e) \parallel p_\theta(Z_e|e)) - E_{q_\phi(Z_e|X,e)}[\mathrm{KL}(q_\phi(Z_{\mathrm{inv}}|Z_e,X) \parallel p_\theta(Z_{\mathrm{inv}}))]$$

It's important to recall that $e$ isn't a random variable, but an index indicating a change in environment - meaning a distribution shift. Therefore, for each such distribution, a different set of parameters is used, leading each probability function conditioned on $e$ to use a distinct set of parameters unique to that environment. In order to differentiate between environment specific parameters and invariant parameters, the environmental parameters will have a subscript $e$ while the causal ones will have subscript $\mathrm{inv}$

$$E_{q_{\phi_e}(Z_e|X)}[E_{q_{\phi_{\mathrm{inv}}}(Z_{\mathrm{inv}}|Z_e,X)}[\log p_{\theta_{\mathrm{inv}}}(X|Z_{\mathrm{inv}},Z_e)]]$$

$$- \mathrm{KL}(q_{\phi_e}(Z_e|X) \parallel p_{\theta_e}(Z_e)) - E_{q_{\phi_e}(Z_e|X)}[\mathrm{KL}(q_{\phi_{\mathrm{inv}}}(Z_{\mathrm{inv}}|Z_e,X) \parallel p_{\theta_{\mathrm{inv}}}(Z_{\mathrm{inv}}))]$$

Assume that everything is Gaussian distributed as a function of the given variables, meaning that

$$p_{\theta_{\mathrm{inv}}}(X|Z_{\mathrm{inv}},Z_e) = \mathcal{N}(\mu_{\mathrm{inv}}(Z), \Sigma_{\mathrm{inv}}(Z))$$

$$q_{\phi_{\mathrm{inv}}}(Z_{\mathrm{inv}}|Z_e,X) = \mathcal{N}(\mu_{\mathrm{inv}}(X,Z_e), \Sigma_{\mathrm{inv}}(X,Z_e)), \quad p_{\theta_{\mathrm{inv}}}(Z_{\mathrm{inv}}) = \mathcal{N}(0,I)$$

$$q_{\phi_e}(Z_e|X) = \mathcal{N}(\mu_e(X), \Sigma_e(X)), \quad p_{\theta_e}(Z_e) = \mathcal{N}(\mu_e(e),I) \quad \forall e \in \mathcal{E}$$

Resulting in

$$\mathrm{KL}(q_{\phi_e}(Z_e|X) \parallel p_{\theta_e}(Z_e)) = E_{q_{\phi_e}(Z_e|X)}\left[\log \frac{q_{\phi_e}(Z_e|X)}{p_{\theta_e}(Z_e)}\right]$$

$$= E_{q_{\phi_e}(Z_e|X)}\left[\log \frac{(2\pi)^{-d/2}|\Sigma_e(X)|^{-1/2}e^{-\frac{1}{2}(z-\mu_e(X))^\top \Sigma_e(X)^{-1}(z-\mu_e(X))}}{(2\pi)^{-d/2}e^{-\frac{1}{2}(z-\mu_e(e))^\top(z-\mu_e(e))}}\right]$$

Using the common assumption that the covariance matrices are diagonal, we get

$$= \frac{1}{2} \sum_{i=1}^{d_e} \left[\Sigma_e(X)_{ii} + (\mu_e(X)_i - \mu_e(e)_i)^2 - 1 - \log \Sigma_e(X)_{ii}\right]$$

Similarly, the invariant KL term can be written as

$$E_{q_{\phi_e}(Z_e|X)}[\mathrm{KL}(q_{\phi_{\mathrm{inv}}}(Z_{\mathrm{inv}}|Z_e,X) \parallel p_{\theta_{\mathrm{inv}}}(Z_{\mathrm{inv}}))]$$

$$= E_{q_{\phi_e}(Z_e|X)}\left[\frac{1}{2} \sum_{i=1}^{d_{\mathrm{inv}}} \left[\Sigma_{\mathrm{inv}}(X,Z_e)_{ii} + \mu_{\mathrm{inv}}(X,Z_e)_i^2 - 1 - \log \Sigma_{\mathrm{inv}}(X,Z_e)_{ii}\right]\right]$$

The expectation $E_{q_{\phi_e}}$ can be approximated using a sampled data point, analogous to the procedure in stochastic gradient descent

$$\Sigma(X) = \begin{bmatrix} \Sigma_{\text{inv}}(X, Z_e(X)) & 0 \\ 0 & \Sigma_e(X) \end{bmatrix}$$

$$\mu(X) = \begin{bmatrix} \mu_{\text{inv}}(X, Z_e(X)) \\ \mu_e(X) \end{bmatrix}$$

$$\mu(e) = \begin{bmatrix} 0 \\ \mu_e(e) \end{bmatrix}$$

The resulting total KL term is simply

$$\text{KL}(q_\phi(Z_e|X, e) \parallel p_\theta(Z_e|e)) + E_{q_\phi(Z_e|X,e)}[\text{KL}(q_\phi(Z_{\text{inv}}|Z_e, X) \parallel p_\theta(Z_{\text{inv}}))] =$$

$$\frac{1}{2} \sum_{i=1}^{d_e + d_{\text{inv}}} \left[ \Sigma(X)_{ii} + (\mu(X)_i - \mu(e)_i)^2 - 1 - \log \Sigma(X)_{ii} \right]$$

## C  ENVIRONMENT TRANSFER ANALYSIS

### C.1  PRELIMINARIES

Let us return to the linear case. Assume the data generating process

$$X_e = \mu_e(e) + A_{\text{inv}} Z_{inv} + A_e Z_e + \epsilon.$$

Where

$$X_e \in \mathbb{R}^D$$
$$Z_{\text{inv}} \in \mathbb{R}^{d_{\text{inv}}}, \quad Z_{\text{inv}} \sim \mathcal{N}(0, I)$$
$$Z_e \in \mathbb{R}^{d_e}, \quad Z_e \sim \mathcal{N}(0, I\sigma_e^2(e))$$
$$\epsilon \sim \mathcal{N}(0, I\sigma_\epsilon^2) \quad \text{Gaussain noise}$$
$$\text{From the causality graph: } Z_{\text{inv}} \perp\!\!\!\perp Z_e, \quad A_i = A_j \;\; \forall i, j \in \mathcal{E}$$

Using this model, we would like to find a closed-form analytical expressions for the encoders and the decoder operations under linearity constraints. The environmental encoder, invariant encoder and decoder weights are obtained by maximizing the probabilities $P^e(Z_e|X_e)$, $P(Z_{\text{inv}}|Z_e, X_e)$, $P(X_e|Z_{\text{inv}}, Z_e)$ respectively. In the linear Gaussian case, this objective reduces to minimizing the mean squared error (MSE), for which the optimal estimators are the corresponding conditional expectations-

$$E_{P^e}(Z_e|X_e), \;\; E(Z_{\text{inv}}|Z_e, X_e), \;\; E(X_e|Z_{\text{inv}}, Z_e).$$

Under the linearity assumption, the conditional expectation estimator is the Wiener estimator of the form:

$$E[A|B] = E[A] + \text{COV}(A, B)\text{VAR}^{-1}(B)(B - E[B])$$

Let us derive it for our model.
The **environmental encoder** is $E_{P^e}[Z_e|X_e]$. The different components are:

$$E[Z_e] = 0$$
$$E[X_e] = \mu_e(e)$$
$$\text{COV}[Z_e, X_e] = \sigma_e^2(e)A_e^\top$$
$$\text{VAR}(X_e) = A_{\text{inv}}A_{\text{inv}}^\top + \sigma_e^2(e)A_e A_e^\top + \sigma_\epsilon^2 I$$

Leading to

$$\hat{Z}_e = E_{P^e}[Z_e|X_e] = \sigma_e^2(e)A_e^\top \left( A_{\text{inv}}A_{\text{inv}}^\top + \sigma_e^2(e)A_e A_e^\top + \sigma_\epsilon^2 I \right)^{-1} (X_e - \mu_e(e))$$

It's important to mention that the variance is invertible only due to the noise, as the other matrices which compose it are of smaller rank. If we assume zero noise and/or want to avoid inverting this close-to-singularity matrix, a computational method should be used instead of an analytic one.

The invariant encoder is $E[Z_{\text{inv}}|X_e, Z_e]$. The different components are:

$$E[Z_{\text{inv}}] = 0$$

$$E\left[\begin{bmatrix} X_e \\ Z_e \end{bmatrix}\right] = \begin{bmatrix} \mu_e(e) \\ 0 \end{bmatrix}$$

$$\text{COV}\left[Z_{\text{inv}}, \begin{bmatrix} X_e \\ Z_e \end{bmatrix}\right] = [A_{\text{inv}}^\top,\ 0]$$

$$\text{VAR}\left(\begin{bmatrix} X_e \\ Z_e \end{bmatrix}\right) = \begin{bmatrix} A_{\text{inv}}A_{\text{inv}}^\top + \sigma_e^2(e)A_eA_e^\top + \sigma_\epsilon^2 I, & \sigma_e^2(e)A_e \\ \sigma_e^2(e)A_e^\top, & \sigma_e^2(e)I \end{bmatrix}$$

Using the fact that the block matrices on the diagonal of the variance matrix are invertible (again, thanks to the noise), we can invert the variance matrix using block-wise inversion formula and get

$$\begin{bmatrix} A_{\text{inv}}A_{\text{inv}}^\top + \sigma_e^2(e)A_eA_e^\top + \sigma_\epsilon^2 I, & \sigma_e^2(e)A_e \\ \sigma_e^2(e)A_e^\top, & \sigma_e^2(e)I \end{bmatrix}^{-1}$$

$$= \begin{bmatrix} (A_{\text{inv}}A_{\text{inv}}^\top + \sigma_e^2(e)A_eA_e^\top + \sigma_\epsilon^2 I - \sigma_e^2(e)A_eA_e^\top)^{-1}, & -(A_{\text{inv}}A_{\text{inv}}^\top + \sigma_e^2(e)A_eA_e^\top + \sigma_\epsilon^2 I - \sigma_e^2(e)A_eA_e^\top)^{-1}A_e \\ -A_e^\top(A_{\text{inv}}A_{\text{inv}}^\top + \sigma_e^2(e)A_eA_e^\top + \sigma_\epsilon^2 I - \sigma_e^2(e)A_eA_e^\top)^{-1}, & \sigma_e^2(e)I \end{bmatrix}$$

$$= \begin{bmatrix} (A_{\text{inv}}A_{\text{inv}}^\top + \sigma_\epsilon^2 I)^{-1}, & -(A_{\text{inv}}A_{\text{inv}}^\top + \sigma_\epsilon^2 I)^{-1}A_e \\ -A_e^\top(A_{\text{inv}}A_{\text{inv}}^\top + \sigma_\epsilon^2 I)^{-1}, & \sigma_e^2(e)I \end{bmatrix}$$

Which yields

$$\hat{Z}_{\text{inv}} = E[Z_{\text{inv}}|X_e, Z_e] = A_{\text{inv}}^\top(A_{\text{inv}}A_{\text{inv}}^\top + \sigma_\epsilon^2 I)^{-1}\left(X_e - \mu_e(e) - A_e Z_e\right).$$

Observe that the invariant encoder turned out to be independent of $\sigma_e^2(e)$.

The **decoder** function is $E[Z_e|X_e]$. The different components are:

$$E[X_e] = \mu_e(e)$$

$$E\left[\begin{bmatrix} Z_{\text{inv}} \\ Z_e \end{bmatrix}\right] = 0$$

$$\text{COV}\left[X_e, \begin{bmatrix} Z_{\text{inv}} \\ Z_e \end{bmatrix}\right] = [A_{\text{inv}},\ \sigma_e^2(e)A_e]$$

$$\text{VAR}\left(\begin{bmatrix} Z_{\text{inv}} \\ Z_e \end{bmatrix}\right) = \begin{bmatrix} I, & 0 \\ 0, & \sigma_e^2(e)I \end{bmatrix}$$

We can easily invert the variance and get

$$\begin{bmatrix} I, & 0 \\ 0, & \sigma_e^2(e)I \end{bmatrix}^{-1} = \begin{bmatrix} I, & 0 \\ 0, & \sigma_e^{-2}(e)I \end{bmatrix}$$

and we finally get

$$\hat{X}_e = E[X_e|Z_{\text{inv}}, Z_e] = \mu_e(e) + A_{\text{inv}}Z_{\text{inv}} + A_e Z_e$$

That is exactly the generative process (minus the noise), which is a nice and intuitive result.

After obtaining analytical expressions for each of the functions of the autoencoder components, the next step is to analyze the specific cases for transferring between seen environments and transferring from unseen environments.

## C.2 Environment Transfer for Seen Environments

For seen environments, the environmental encoder is that of the source environment $e = e_s \in \mathcal{E}_{\text{train}}$:

$$\hat{Z}_{e_s} = E_{P^{e_s}}[Z_e|X_e] = \sigma_e^2(e_s)A_e^\top\left(A_{\text{inv}}A_{\text{inv}}^\top + \sigma_e^2(e_s)A_eA_e^\top + \sigma_\epsilon^2 I\right)^{-1}(X_e - \mu_e(e_s))$$

The invariant encoder is directly dependent on $e_s$ solely through the centering of $X_e$ (the mean term):

$$\hat{Z}_{\text{inv}} = E[Z_{\text{inv}}|X_{e_s}, Z_{e_s}] = A_{\text{inv}}^\top(A_{\text{inv}}A_{\text{inv}}^\top + \sigma_\epsilon^2 I)^{-1}(X_{e_s} - \mu_e(e_s) - A_e Z_{e_s})$$

Finally, the decoder uses an environmental component from the target environment:

$$\hat{X}_{e_t} = E[X_e|Z_{\text{inv}}, Z_{e_t}] = \mu_e(e_t) + A_{\text{inv}}Z_{\text{inv}} + A_e Z_{e_t}$$

## C.3 Environment Transfer for Unseen Environments

For $e_s \in \mathcal{E}_{\text{test}}$, we assume that $A_e, A_{\text{inv}}, \sigma_\epsilon^2$ and $\mu_e(e), \sigma_e^2(e) \ \forall e \in \mathcal{E}_{\text{train}}$ are known, and $\mu_e(e_s), \sigma_e^2(e_s), \ e_s \in \mathcal{E}_{\text{test}}$ are unknown. This requires us to derive the environmental encoder again $\hat{Z}_{e_s} = E_{P^{e_s}}[Z_e|X_e]$.

$$
\begin{aligned}
E[Z_{e_s}] &= 0 \\
E[X_{e_s}] &= \mu_e(e_s) : \text{ unkown} \\
\text{COV}[Z_{e_s}, X_{e_s}] &= \sigma_e^2(e_s)A_e^\top : \ \sigma_e^2(e_s) \text{ unkown} \\
\text{VAR}(X_{e_s}) &= A_{\text{inv}}A_{\text{inv}}^\top + \sigma_e^2(e)A_eA_e^\top + \sigma_\epsilon^2 I : \ \sigma_e^2(e_s) \text{ unkown}
\end{aligned}
$$

### C.3.1 Heuristic Motivation

Let us give a quick motivation for our heuristic method for unseen environment transfer. For unseen environment transfer, we're trying to estimate an "equivalent" target data point from a seen environment $X_{e_t}, \ e_t \in \mathcal{E}_{\text{train}}$ to a data point from an unseen environment $X_{e_s}, \ e_s \in \mathcal{E}_{\text{test}}$, using the data we gathered from our training environments. We can therefore consider the following MSE optimization problem

$$
\hat{X}_{e_t} = \arg\min_{X_{e_t}^*} \sum_{e \in \mathcal{E}_{\text{train}}} \mathbb{E}_{P^e} \left[ ||X_e - X_{e_t}^*||_2^2 | X_{e_s} \right].
$$

The minimum is obtained by differentiating and locating a stationary point

$$
\sum_{e \in \mathcal{E}_{\text{train}}} (2\hat{X}_{e_t} - 2\mathbb{E}_{P^e}[X_e|X_{e_s}]) = 0,
$$

and the resulting estimation is

$$
\hat{X}_{e_t} = \frac{1}{|\mathcal{E}_{\text{train}}|} \sum_{e \in \mathcal{E}_{\text{train}}} \mathbb{E}_{P^e}[X_e|X_{e_s}].
$$

This derivation can also be repeated for $Z_{e_s}$ instead of $X_{e_t}$, that way, we can derive an expression for the environmental encoder.

For $Z_{e_s}$ we get that

$$
\begin{aligned}
\hat{Z}_{e_s} &= \arg\min_{Z_{e_s}^*} \sum_{e \in \mathcal{E}_{\text{train}}} \mathbb{E}_{P^e} \left[ ||Z_e - Z_{e_s}^*||_2^2 | X_{e_s} \right] \\
\hat{Z}_{e_s} &= \frac{1}{|\mathcal{E}_{\text{train}}|} \sum_{e \in \mathcal{E}_{\text{train}}} \mathbb{E}_{P^e}[Z_e|X_{e_s}] \\
&= \frac{1}{|\mathcal{E}_{\text{train}}|} \sum_{e \in \mathcal{E}_{\text{train}}} \sigma_e^2(e)A_e^\top \left( A_{\text{inv}}A_{\text{inv}}^\top + \sigma_e^2(e)A_eA_e^\top + \sigma_\epsilon^2 I \right)^{-1} (X_{e_s} - \mu_e(e)).
\end{aligned}
$$

We use this result as motivation behind our heuristic estimation function of $Z_{e_s}$ for unseen source environment, which is

$$
\hat{Z}_{e_s} = \frac{1}{|\mathcal{E}_{\text{train}}|} \sum_{e \in \mathcal{E}_{\text{train}}} Z_e \quad ; \quad Z_e \sim P^e(\cdot|X_{e_s}).
$$

## D Applications to Fairness

Until now, we have examined the IRM framework primarily through the lens of robustness to distribution shifts across environments. However, previous works, such as Hardt et al. (2016) demonstrate that similar approaches can be effectively applied in the context of algorithmic fairness between subpopulations. In this view, the environmental features correspond to sensitive attributes such as race, gender, or age, variables we would like the model to be invariant to. This perspective is especially relevant in socially sensitive applications such as hiring, admissions or lending, where it is essential

to ensure that decisions are made based on meritocratic and relevant factors (e.g., qualifications, experience, or financial stability), rather than irrelevant or discriminatory ones. Extensive research has been devoted to learning fair representations, underscoring the significance of this problem. Notable contributions include Zemel et al. (2013), Creager et al. (2019) and Locatello et al. (2019).

Within our framework, we can interpret these *relevant* factors as *invariant features*, and the sensitive or discriminatory attributes as *environmental features*. In this section, we explore the potential of the VIAE algorithm for promoting fairness by separating invariant (relevant) and environment-specific (irrelevant) components in the learned representation.
We use the architecture suggested in Subramanian (2020) for the decoder and invariant encoder, while using a reduced encoder variant for each environmental encoder.

## D.1 CelebA Dataset

To demonstrate VIAE in the context of fairness, we use the CelebA dataset (Liu et al., 2015), which contains over $200,000$ face images of celebrities annotated with 40 binary attributes. We focus on the "Male" attribute to define two subpopulations: "male" and "female". In our setup, this attribute serves as the environmental variable, representing a sensitive feature we aim to disentangle from invariant content. This is a natural choice, as gender-related biases in facial recognition and classification systems can be a source of significant ethical concerns in real-world applications. Our goal is to encompass the gender-related characteristic of the CelebA images in the environmental part of the latent space, while the invariant part contains the other, non gender specific features. Under this setting, we expect VIAE to generate gender-specific reconstructions conditioned on the appropriate prior, enabling independent control over invariant and environment-specific aspects of the generated samples. Furthermore, VIAE allows for *environment transfer*: converting a given image from one subpopulation (gender) to the other, while preserving the invariant features such as facial structure, expression, pose and so on. This setup provides a concrete use case for testing the fairness-promoting capacity of VIAE: a model that can modify sensitive attributes without affecting identity-relevant features can preclude discriminatory prediction biases and promote fairness.

## D.2 Sample Generation

Following the approach used for SCMNIST and SMNIST datasets, Figure 7 illustrates the generation behavior of VIAE on the CelebA dataset. Similarly to the SMNIST/SCMNIST cases, we draw a single sample of the invariant features $Z_{\mathrm{inv}}$ and generate images using five different samples from each of the two environment-specific priors. The model successfully identifies the unique characteristics of each subpopulation, corresponding to the "male" and "female" environments. It produces coherent samples for each group, with the top five images generated from the "male" environment prior and the bottom five from the "female" prior. Notably, we can notice a similarity between the different samples, suggesting that they share some set of features which the model learned to be "invariant to gender" in one way or another.

## D.3 Environment Transfer

We further evaluate VIAE's ability to perform environment transfer between seen environments. Specifically, we change the environmental component of the latent space to transfer images from the "male" domain to the "female" domain, while keeping the invariant representation fixed. As shown in Figure 8, the algorithm successfully transforms "male" images into "female" counterparts. Importantly, several identity-related features—such as facial structure, expression, and pose—remain somewhat consistent between the original and transferred images, indicating that these attributes are captured in the invariant representation. We emphasize that this experiment is not intended to compete with current state-of-the-art generative models in terms of visual fidelity or realism. Rather, it serves as a qualitative validation of VIAE's capacity to disentangle and manipulate environmental features independently. These preliminary results highlight the potential of the VIAE framework for controlled generation and fairness applications, motivating future work to explore more advanced architectures and training strategies.

Environment 1

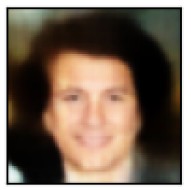 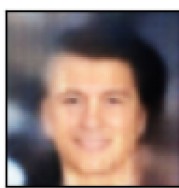 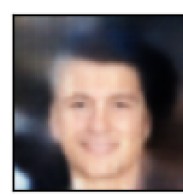 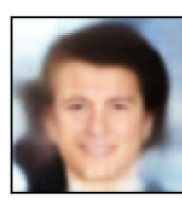 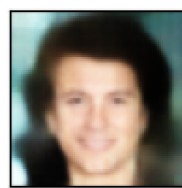

Environment 2

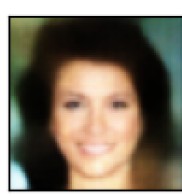 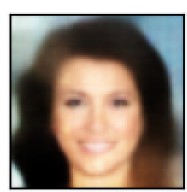 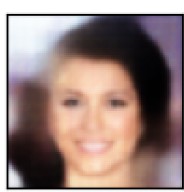 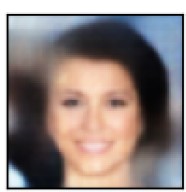 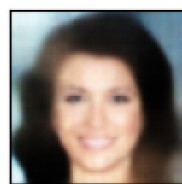

Figure 7: CelebA sample generation. The top five images were created using five samples from $Z_e \sim \mathcal{N}(\mu_e(1), I)$ (first environment, "male"), and the bottom five using five samples from $Z_e \sim \mathcal{N}(\mu_e(2), I)$ (second environment, "female"), while keeping $Z_{\text{inv}}$ fixed.

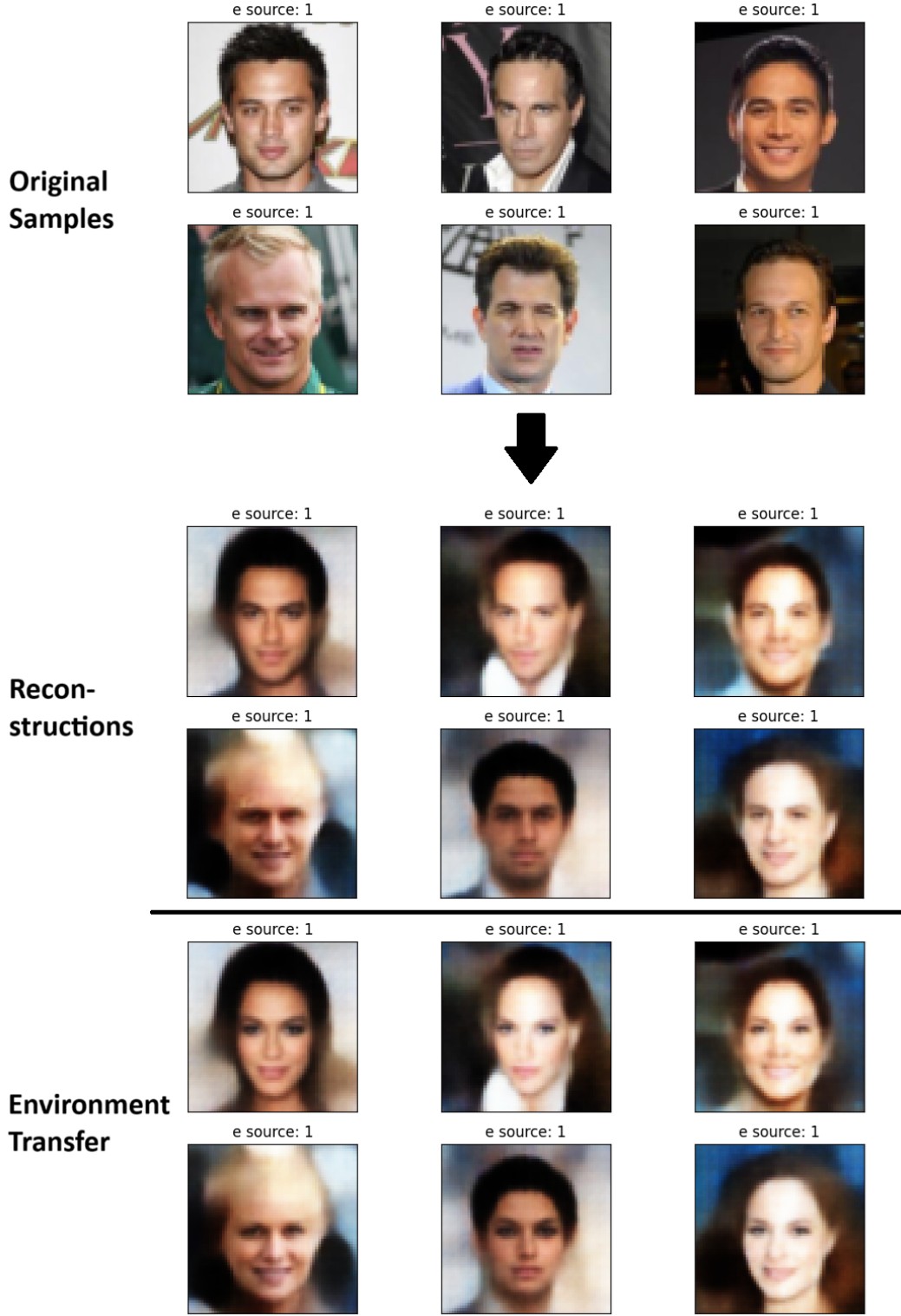

Figure 8: Environment transfer from the "male" subpopulation to the "female" subpopulation. The top two rows show original images $X_{e_s}$, the next two rows show their respective reconstructions $\hat{X}_{e_s}$ and the last two rows show their respective transferred versions $\hat{X}_{e_t}$.

