# OpenReview forum: "Unsupervised Representation Learning - an Invariant Risk Minimization Perspective"
_ICLR.cc/2026/Conference — ICLR 2026 Poster_

### Official Review · Reviewer_XTfo · 2025-10-16

**Soundness:** 3
**Presentation:** 3
**Contribution:** 3
**Rating:** 8
**Confidence:** 4

**Summary:**

The paper proposes two unsupervised algorithms inspired by the invariant risk minimization (IRM) paper.  PICA (principal invariant component analysis) is in fact quite constrained and therefore less interesting than VIAE (variational invariant auto-encoder) which works by splitting the latent into an invariant component and an environment dependent component.  VIAE can also be used for environment transfer and to possibly recover IRM

**Strengths:**

- The notion of unsupervised invariant algorithms is compelling
- The VIAE approach seems promising (as illustrated by the environment transfer algorithm.).

**Weaknesses:**

- Some notations are confused (see below)
- PICA is very constrained because the difference of two empirical covariance matrices $\Sigma_1-\Sigma_2$ is likely to have a null kernel.  VIAE is far more satisfactory.
- Experiments in section 4.2.1 are insufficient. For instance, you could construct a CMNIST problem with labels that only depend on the shape of the digit, and a RevCMNIST problem using the same patterns but with labels that only depend on the color. The invariant features in the sense of IRM would be the shape for CMNST and the color for RevCMNIST. But since both datasets have the same input patterns, the unsupervised approach cannot make this difference.

**Questions:**

Imprecise notations:
- line 88. $(X,e)\sim P_X^e(X)$.  If $P^e_X$ is a distribution over $X$, then it does not generate pairs $(X,e)$
- line 91. What's the relation between $P^e_X$ and $P_X$ ?
- line 98. Should there be an additional sum or an empirical average on the data for each environment.
- line 102. If the definition of $P(X,Y)$ implicitly depends on $\Phi$, why write $P(X,\Phi(X))$ in line 98?

Questions:
- What is the relation of PICA and CCA (canonical correlation analysis, Hoteling 1936)?
  Hoteling is also the inventor of PCA btw.
- Isn't PICA very constrained as the difference of two empirical convariance matrices  $\Sigma_1-\Sigma_2$ can easily have a null kernel.
- Line 286. Why not the other direction $P_e(Z_{inv},Z_e|X)=P(Z_{inv}|X) P_e(Z_e|Z_{inv},X)$ ?

---

> ### Author Response · Authors · 2025-11-19
>
> Hello,
>
> We thank the reviewer for the valuable input on our paper.
>
> We would like to address the weaknesses you mentioned:
>
> 1. The notations-related issues will be addressed in the questions answering part below.
>
> 2. PICA is presented in this paper as a linear motivation and for analysis of the unsupervised IRM problem. In practical setup, you are indeed right and instead of looking for the kernel of $Σ_1-Σ_2$, one should look for eigenvectors with corresponding eigenvalues that are lower than some threshold.
>
> 3. We argue that the invariant features remain the shape of the digit in that case, as this is still the property that remains stable across environments in both SCMNIST and RevCMNIST, regardless of the labels. In this particular example, the labeling itself is not invariant, and therefore, our unsupervised framework will indeed be unhelpful for a color prediction downstream supervised task. Moreover, this example is quite similar to the experiment we performed in section 4.3, with environment prediction e ̂ corresponding to RevCMNIST labels, the results in this section support the claim stated here.
>
> We would now like to move on and address your questions:
>
> 1. We agree with this correction and have changed the notation to  $X_e∼P_X^e (X)$.
> 2. We agree that this notation is confusing and have changed it to
> $P_θ^i (ϕ(X))=P_θ^j (ϕ(X))  ∀i,j∈\mathcal{E}_{all} $.
>
> 3. In this equation, $X$ is a generic input variable. When translating this formula to the case with actual data-samples which are used to approximate this formula, you are completely right, and an extra summation over samples is needed.
> 4. We believe that you meant to write $P(X,Z)$. If that is indeed the case, it is important for us to highlight that $Z=Φ(X)$, meaning that it is an embedding/transformation of $X$.
>
> 5. The tasks in CCA and in our present setting are different. CCA deals with random variables X and Y which are often of different dimensions and modalities, and the aim is to find two transformations which render the transformed variables as correlated as possible. Our framework is different, in that $X_{e_1}$ and $X_{e_2}$ have the same data type, and the objective is to extract the common invariant features to both environments, using a single transformation $Φ(⋅)$.
>
> 6. PICA limitations have been addressed in the weaknesses part above.
>
> 7. When we first developed VIAE, we indeed tried to first use $P^e (Z_{inv},Z_e│X)=P(Z_{inv}│X) P^e (Z_e│Z_{inv},X)$, as it posses the favorable property that $Z_{inv}$ can be reconstructed without $Z_e$. The problem with this approach is that what we actually get is  $P^e (Z_{inv},Z_e│X)=P^e (Z_{inv}│X) P^e (Z_e│Z_{inv},X)$, as the collider property for causality graphs does not allow us to determine that $Z_{inv}$ is independent of the environment once $X$ is given. Alternatively, when using $P^e (Z_{inv},Z_e│X)=P^e (Z_e│X) P^e (Z_{inv}│Z_e,X)$, the dependence on the environment in the $P^e (Z_{inv}│Z_e,X)$ term is alleviated due to the chain (Markov) property, allowing us to have a single invariant encoder instead of a different one for each environment.
>
> Thank you again for the thorough review and important remarks and clarifications, we hope we have addressed your concerns in a satisfactory manner. We will be happy to address any further questions and concerns you might have.

---

### Official Review · Reviewer_wMwF · 2025-10-28

**Soundness:** 2
**Presentation:** 3
**Contribution:** 1
**Rating:** 2
**Confidence:** 4

**Summary:**

This paper proposes unsupervised Invariant Risk Minimization (IRM), extending the IRM framework to settings without labels by redefining invariance through feature distribution alignment across environments. The authors introduce two methods: Principal Invariant Component Analysis (PICA), a linear Gaussian approach that identifies invariant directions via null-space projections, and Variational Invariant Autoencoder (VIAE), a deep generative model that factorizes latent representations into environment-invariant (Z_inv) and environment-specific (Z_e) components. The framework is evaluated on synthetic data, modified MNIST variants (SMNIST, SCMNIST), and preliminary experiments on CelebA.

**Strengths:**

- The problem formulation--extending IRM to unlabeled multi-environment data--appears to be novel.
- The two-method approach (linear PICA + nonlinear VIAE) provides complementary perspectives with clear mathematical exposition.
- The fairness application demonstrates a natural use case where environment-invariant features correspond to removing sensitive attributes.

**Weaknesses:**

- One of the core weaknesses is that the objective is conceptually ill-defined. The paper redefines invariance as matching the marginals of the representations across environments but does not justify how this invariance serves IRM's goal of robust prediction. For instance, the learnt invariant features could be useless for any feasible downstream tasks.
- Connecting to the previous point, there is no identifiability analysis of the learned representation; the model could learn arbitrary rotations of the invariant features [see 1] and there is no theoretical or empirical justification along those lines. I am also unsure if the learned representation would be coherent in all cases, for instance when data distribution is unbalanced across environments (as an extreme case, consider all 3's in one environment, all 1's in another).
- The paper is missing extremely relevant prior literature on learning disentangled representations [1, 2, 3, 4] and fairness via disentanglement [11, 12, 13], as well as less critical but appropriate citations works on causal representation learning, IRM and domain generalization [5, 6, 7, 8, 9, 10], among others. Similarly, links between section 4.2 and references in domain adaptation should be fleshed out.
- Empirical justification is very limited: (1) MNIST-based simple datasets, no baselines - zero comparison to disentangled representations such as $ \beta $-VAE, or supervised IRM methods (2) No quantitative disentanglement metrics, evaluation relies on visual inspection; (3) Section 4.3's claim that 84% linear probe accuracy "validates" the approach lacks context without supervised baselines showing achievable performance.

Overall: The paper explores an interesting question but suffers from fundamental conceptual ambiguity (what is "invariance" without labels?) and insufficient empirical rigor (simplistic datasets, no baselines, no disentanglement metrics, no supervised comparisons).

References:

[1] Challenging Common Assumptions in the Unsupervised Learning of Disentangled Representations - Locatello et al. 2019
[2] beta-VAE: Learning Basic Visual Concepts with a Constrained Variational Framework - Higgins et al., 2017
[3] Isolating Sources of Disentanglement in Variational Autoencoders - Chen et al, 2018
[4] Disentangling by Factorising - Kim et al, 2018
[5] Towards Causal Representation Learning - Schölkopf et al., 2021
[6] Weakly Supervised Disentangled Generative Causal Representation Learning - Shen et al, 2022
[7] On Learning Invariant Representations for Domain Adaptation - Zhao et al, 2019
[8] Distributionally Robust Neural Networks for Group Shifts: On the Importance of Regularization for Worst-Case Generalization - Sagawa et al, 2020
[9] Learning Optimal Features via Partial Invariance - Choraria et al., 2023
[10] Context is Environment - Gupta et al, 2023
[11] Learning Fair Representations - Zemel et al., 2013
[12] Flexibly Fair Representation Learning by Disentanglement - Creager et al., 2019
[13] On the Fairness of Disentangled Representations - Locatello et al., 2019

**Questions:**

- Can the authors provide some grounded justification for distributional equivalence?
- Can the authors demonstrate a practical use-case, backed with results, of the presented method?

---

> ### Author Response · Authors · 2025-11-19
>
> Hello,
>
> We thank the reviewer for the valuable input on our paper.
>
> We would like to address the weaknesses you mentioned:
>
> 1. We argue that any unsupervised framework is prone to be “ill-defined” in this sense (see, for example, Ben-David, Lu, and Pál. "Does Unlabeled Data Provably Help? Worst-case Analysis of the Sample Complexity of Semi-Supervised Learning." COLT. 2008).
>
> When considering an unsupervised framework, whether it is PCA, K-means, VAE or PICA/VIAE, the objective is to learn some underlying structure of the unlabeled data, in the hope that this structure contains information which could be useful in some sense. However, the ‘sense’ in which the representation is useful depends very much on future tasks, which are, by definition, unknown.
> To the best of our knowledge, unsupervised frameworks cannot guarantee performance on an arbitrary downstream task (see above reference to Ben-David paper), this limitation applies to our work as well, and yet, we think that there’s still value in considering the framework we offer.
> In Section 4.2 we introduce the “Environment Transfer” scheme, which ideally can turn a dataset that is divided between different environments to an equivalent dataset which contains a single environment. This is our alternative objective, corresponding to supervised IRM’s robust predictor. We argue that eliminating distribution shifts in the dataset can be a satisfying solution to the unsupervised IRM problem, which we define in this paper and is inherently different from the supervised IRM case.
>
> 2. Since our framework is an unsupervised one, the model cannot reason about labels (in the MNIST example, the digits), all it can do is try and separate the environment-invariant features from the environment-dependent ones. In the examples we provided, we specifically engineered the MNIST-based datasets so that the digits are the invariant features, and therefore, as was shown in the paper- these properties were indeed the ones that were captured by the invariant encoder. In the example you provided, digits are not invariant, and therefore, since our model has no access to labels- there’s no reason for it to treat the “digit shape” property as invariant.
>
> 3. It is important to emphasize that our work focus is not on learning disentangled representations. Our only goal is to separate the environment-invariant features from the environment-dependent ones. We do not claim to achieve disentanglement between the different elements of either the invariant representation or the environmental one.
> Concerning the fairness representation learning works you addressed, these are indeed very relevant to the “Application for Fairness” section, and we’ve addressed them in the paper revision (Section D, starting at line 973). We thank you for bringing these papers into our attention.
> As for the causal representation learning papers, we have addressed some of them in section 1.3 (line 123), we thank you again for bringing these into our attention.
> Finally, in section 4.2 we have addressed the major works in domain adaptation:
>
> $\bullet$ “Adversarial discriminative domain adaptation” by Eric Tzeng, Judy Hoffman, Kate Saenko, and Trevor Darrell.
>
> $\bullet$ “Unpaired image-to-image translation using cycle-consistent adversarial networks” by Jun-Yan Zhu, Taesung Park, Phillip Isola, and Alexei A. Efros.
>
> We would be happy to address more papers we might have wrongfully overlooked if needed.
>
> 4. We would like to once again emphasize that this work does not deal with learning disentangled representations, and therefore, it cannot be compared against disentanglement schemes such as beta VAE and similar.
> The purpose of the demonstration in section 4.3 is twofold. First, we wanted to show that the learned representation does indeed contain the information “it was supposed to”, i.e the invariant representation learned the invariant information (digits shape) and the environmental representation learned the environmental information (squares locations and digits colors), as demonstrated by the prediction score for the labels and environment labels accordingly. The second, and perhaps more important thing we wanted to show, is that the invariant features perform poorly on the environment prediction task. This demonstration shows that the invariant features do not contain environmental information, as desired in the general supervised IRM framework.
> Finally, we emphasize that we do not compare ourselves numerically to other works since this is a new problem setup, which we believe is interesting and is worth further investigation.
>
> As for the final remark, we believe that invariance in the unsupervised settings can be concisely described as the latent features which do not depend on the environment.
>
> We will address your questions in the next comment due to limited space.

---

> ### Author Response · Authors · 2025-11-19
> **Questions Answering**
>
> We would now like to move on and address your questions:
>
> 1. We believe that by “distributional equivalence” you mean that
> $P_θ^{e_1 } (ϕ(X))=P_θ^{e_2 } (ϕ(X))$. For the linear Gaussian case, we have provided an analysis (section A) which shows that a linear transformation can extract the subspace of the invariant features, resulting in distributional equivalence of the (linear) embedding for different environments. In the general setting we seek a nonlinear mapping  $ϕ(X)$ such that the distributions of the embeddings are equal, generalizing the linear case where distributional equivalence in subspaces suffices.  For this case, the resulting qualitative results in our ability to generate samples with the same apparent (nonlinear) invariant features (digits shape for MNIST-based datasets, facial features that are non-related to sex for CelebA), together with the quantitative results of section 4.3, which shows that the invariant features performs poorly when used to predict the environment, support our claim for distributional equivalence.
>
> 2. The demonstration in section D: “Application for Fairness” shows how this method can be used for domain adaptation. Also, we believe that the concept of “Environment Transfer” can be used to alleviate some cases of out-of-distribution learning problems, which is of great use to the community.
>
> Thank you again for the thorough review and important remarks and clarifications, we hope we have addressed your concerns in a satisfactory manner. We will be happy to address any further questions and concerns you might have.

---

> ### Comment · Reviewer_wMwF · 2025-11-20
> **Response to Rebuttal**
>
> I thank the authors for the rebuttal, but my concerns are still not addressed.
>
> - Regarding "ill-defined": My issue is not with the unsupervised nature of representation learning, but with the constraint on the feature space i.e. $P_θ^{e_1 } (ϕ(X))=P_θ^{e_2 } (ϕ(X))$. If invariance is only specified with respect to the environment, then the learnt features might be completely useless for any "risk-related" downstream tasks and hence, the term "risk minimization" does not make sense.
>
> - The reason why "disentangled representations" are extremely relevant is because the environment can now simply be treated as a latent that varies across data, and such a disentanglement of it (and corresponding domain transformation) is demonstrably feasible with something like $\beta$-VAE (see their results with gender on CelebA). Not to mention the plethora of works that have followed prior/since.
>
> - When analyzing this work from a domain invariance point of view (i.e. learning a representation that's invariant across all domains/environments), the work has very limited novelty. Specifically, PICA has already appeared almost identically in [1], in the the context of invariant feature subspace recovery (see the method corresponding to ISRCov in [1]). Similarly, the concept of learning an invariant representation that can be transferred to multiple domains has been explored with a very similar motivation in [2]. While their eventual goal is downstream prediction, the elicited representation at the intermediate steps yields either exactly the same properties, or very similar ones that this work seems to claim. The corresponding empirical validation is also much better fleshed out in the corresponding works.
>
> - Therefore, this work's contribution becomes largely methodological (as stated by reviewer vC27) i.e. how "good" is the invariant representation achieved by the proposed optimization scheme. And without any quantitative evaluation of the quality of the representation via downstream tasks on real datasets beyond MNIST or practical applications, and without any comparison to baselines that can be employed to elicit similarly invariant representations, even this contribution cannot be ascertained. Even framed as 'environment transfer,' this is essentially domain adaptative transfer, and comparisons to domain adaptation methods are still necessary.
>
> - Finally, with regards to section 4.3, the performance with the learned representation (84% on numbers, 58% on environment) are hard to interpret without any comparison to what a supervised baseline can achieve. A simple sanity check to put these numbers into context, is repeating the same with the representation learnt by IRM. This at least allows to quantify the gap due to having less information in the unsupervised case as well as the effective level of invariance compared to IRM, meaningfully.
>
> References
>
> [1]:  Provable Domain Generalization via Invariant-Feature Subspace Recovery - Wang et al, 2022.
>
> [2]: Domain Invariant Representation Learning with Domain Density Transformations - Nguyen et al., 2022

---

> ### Author Response · Authors · 2025-11-24
>
> We thank you for the follow-up.
>
> 1. We think it’s important to emphasize that the constraint $P_θ^{e_1 } (ϕ(X))=P_θ^{e_2 } (ϕ(X))$ is not the full story in our setting. The full objective, as can be seen in equation 2, is to maximize the sum of log likelihoods over the environments ($logP^e (X,ϕ(X))$). This objective pushes the model to learn meaningful representations, under the constraint that they are invariant.
> If the reviewers find it problematic to address a log-likelihood (or minus log likelihood for this matter) as risk, we are open to changing the name of the paper to something along the lines of “Unsupervised Representation Learning - an Invariant Risk Minimization perspective” or similar.
>
> 2. We emphasize that our objective is inherently different than that of beta VAE. In beta VAE, the authors aim to achieve disentanglement between the different elements of their learned representation by giving a higher weight to the KL divergence term, which pushes the learned representation closer towards the standard normal distribution. This approach is prone to the limitations pointed out by [1]. Our method is fundamentally different, we make use of environment “labels” to learn two separate representations, one that is invariant across different environments and one that is not. It is important to emphasize that we do not claim to achieve a disentangled invariant representation (or a disentangled environmental representation for that matter), the different elements which compose the invariant representation can be arbitrarily entangled. By utilizing the environment “labels”, we can enforce the inductive bias of our VIAE causally inspired architecture, which allows us to avoid the pitfall described in [1].
>
> 3. We do not claim our work is the first to introduce domain adaptation (line 368). Our work offers a new method, specifically designed with the IRM framework in mind. Several points are important to note regarding the differences between our work and [3]: (1) It offers generative capabilities ([3] uses a generative network in the form of GAN, but no generative examples are presented), (2) it explicitly models a separated latent space- which allows us to perform interventions on both the environment (domain) specific features and/or the invariant features, unlike [3]. (3) the authors of [3] specifically claim that they rely on the assumption that the labels are invariant to the domain (environment), this assumption corresponds to the PIIF (partially informative invariant features) case, which we mention in line 127. Our scheme does not need this assumption, and our proposed unsupervised SCM generalizes both FIIF and PIIF cases. It is also important to note that the PIIF case is considered to be the less challenging setup, as shown in [2].
>
> As for ISRCov, it is indeed somewhat similar to PICA, yet part of PICA’s contribution is for motivation and analysis which are carried out throughout the paper. We also use the PICA framework to offer the probabilistic PICA algorithm (section A.3) and as motivation for unseen environment transfer (C.3.1).
>
>
>
> 4. The important comparison to be made in section 4.3 is between the different embeddings. You can see the complementary qualities between the performance of the invariant features and the environmental features. The aim of this work is not to offer a new SOTA algorithm, but to offer a new approach and framework to a known problem.
>
>
> References:
>
>
> [1] Challenging Common Assumptions in the Unsupervised Learning of Disentangled Representations - Locatello et al. 2019
>
> [2] Invariance Principle Meets Information Bottleneck for Out-of-Distribution Generalization – Ahuja et al. 2021
>
> [3] Domain Invariant Representation Learning with Domain Density Transformations - Nguyen et al., 2022

---

> ### Comment · Reviewer_wMwF · 2025-11-26
> **Response to Comment**
>
> - On "risk minimization": The log-likelihood objective is standard for any generative model (VAE, diffusion models, etc.). Simply maximizing likelihood under an invariance constraint does not constitute "risk minimization" in the IRM sense, and the paper would benefit significantly from the reframing the contribution.
> - On disentanglement: You misunderstand my point. The problem of separating environment-dependent factors from environment-independent ones is a disentanglement problem. While the impossibility result holds without any inductive biases, adding environment labels moves it into the realm of weak supervision. This problem has also been addressed before. For instance, Gabbay et al. (2021) [1] demonstrate disentanglement using attributes, which is structurally analogous to the use of environment labels in this work (color in case of MNIST, gender in case of Celeb-A). The "Inductive bias via environment labels" is precisely the weak supervision that enables disentanglement. Mentioning $\beta$-VAE was a starting point intended to spur the authors into doing a proper literature survey in disentanglement and the work that has followed since. These connections must be acknowledged and methods compared.
> - On novelty: Re ISRCov: "Somewhat similar" understates the overlap. The core algorithmic contribution is almost identical to prior work, and probabilistic PICA and heuristic environment transfer are incremental extensions. Regarding the assumption of that "labels are invariant to the domain (environment)", the experiment with CMNIST uses it implicitly. As stated in the earlier rebuttal, "we specifically engineered the MNIST-based datasets so that the digits are the invariant features", and the fact that this method will fail if one were to create unbalanced digit distributions in the environments, is a tacit reaffirmation of this.
> - On validation: "The aim is not SOTA, but a new framework" does not excuse the absence of baselines. To validate a "new framework," one must demonstrate: (a) it can recover a known solutions in supervised settings to an extent, (b) the 84%/58% numbers are meaningful relative to what's achievable. Without supervised IRM comparison, these numbers remain uninterpretable.
>
> The work needs either (1) rigorous repositioning as weakly-supervised disentanglement with proper comparisons to related methods like [1], or (2) clear demonstration of novelty and advantages over existing invariant representation methods. Neither has been provided.
>
>
> References:
> [1] An Image is Worth More Than a Thousand Words: Towards Disentanglement in the Wild - Gabbay et al., 2021

---

> > ### Author Response · Authors · 2025-12-01
> >
> > 1.	“Likelihood” is a standard objective in unsupervised learning. In parallel, “risk” is a standard term for minimization-based objectives, which makes it a natural analog in IRM-style formulations. That being said, since two reviewers found the title confusing, we’ve decided to change it to “Unsupervised Representation Learning - an Invariant Risk Minimization Perspective”. We believe that this name reflects both the background and context in which this work was done, and also eliminates the confusion regarding the overall objective of the paper.
> > 2.	We have agreed with some of the reviewer’s prior suggestions for added citations and related work, and have added them to the “Related Work” section (line 154). However, we believe the “Related Work” section now covers the overall literature context in which this work has been done with adequate framing (containing both IRM and representation learning). We’ve also explained what separates this work from disentanglement-focused papers such as beta VAE. Finally, we believe that the IRM framing we provided provides the required context for what the reviewer calls “weak supervision”.
> > 3.	As in previous comments, the reviewer seems to focus only on the constraint alone rather than on the overall objective. Our general objective (Equation 2) includes both a constraint and an objective, with both being crucial in the IRM framework. Equivalently, PICA involves more than identifying the invariant directions (as done in ISRCov). As done in other IRM methods, PICA maximizes a PCA-style objective under invariance constraints, making it a simple and intuitive unsupervised algorithm for the IRM framework. The probabilistic PICA formulation and the derivation for the environment-transfer are important components that establish the connection between PICA and VIAE, with the environment-transfer derivation also serving as motivation for the heuristic “Unseen Environment Transfer” scheme. We respectfully disagree with the statement that they are merely “incremental extensions”. Regarding the PIIF assumption, since we do not assume access to labels, let alone specific causal relationships involving them, it does not apply in our setting.
> > 4.	We emphasize that section 4.3 should be addressed as an experiment done to test the resulting (unsupervised) embeddings, rather than as a competitive supervised-IRM algorithm suggestion. For example, as stated in the paper (line 476), the most important result is actually the failure of prediction based on the invariant features to identify the original environment, which the reviewer did not mention. What baseline can be used for this comparison? Moreover, what constitutes a more meaningful comparison than pure chance (line 477), to which our result is close?
> >
> > To summarize, our work’s main objective is to offer an unsupervised framework and solutions to the IRM problem, with consideration to its specific characteristics. We believe it’s important to keep that in mind when considering the overall framing, the related work, and the proposed objectives and algorithms.
> >
> > We thank the reviewer again for the continuous feedback and remarks, and hope our reply addresses the concerns raised.

---

### Official Review · Reviewer_vC27 · 2025-10-31

**Soundness:** 3
**Presentation:** 3
**Contribution:** 2
**Rating:** 6
**Confidence:** 4

**Summary:**

The paper proposes an invariance principle for unsupervised learning, where an optimal reconstruction is sought from a latent representation that is constrained to be identically distributed along the training environments. The principle is demonstrated on Gaussian data, then a tailored VAE architecture for the problem is presented and demonstrated on MNIST variants and CelebA.
The empirical results show the model learns latent representations that hold domain invariant features, and another latent representation that holds domain dependent features.

**Strengths:**

Overall I liked the topic of the paper, the discussion presented and the developed methods.
I think there are some original contributions that are presented clearly, and the work could attract some interest from crowd interested in these topics.

**Weaknesses:**

There are some apparent weaknesses that I think the authors should take into account when revising the paper:
1. The motivation for the problem is not entirely clear. "Risk minimization" is a term mostly used in the context of a prediction problem, and I think the unsupervised setting is inherently different, hence a more suitable name for the work or solution might be something like an Invariant autoencoder/Environment-Invariant autoencoder etc. Now given this framing, it is not entirely clear what one has to gain from invariance in the unsupervised case, and what is the spurious correlation present at training time. Are we expecting the reconstruction error to be stable or min-max optimal on new environments, or some other form of robustness? The unclarity about motivation is especially apparent in section 4.2 where the authors link the problem back to supervised learning. I was not fully able to follow the claims in this section, and I think some more conceptual clarity is required either via math (as suggested in point 3 below), or a more convincing empirical problem.
2. The idea seems quite similar to works from the domain adaptation and generalization literature, like DICA, DANN, CORAL, and Domain Separation Networks [1, 2, 3, 4]. Some of them are more closely related than others to the work under review, but it seems important that the authors refer to these works and explain the conceptual differences. Other than the presence of a label, the invariance principle where one wishes to learn a representation $\Phi(X)$ such that $P_{e}(\Phi(X)) = P_{e'}(\Phi(X))$ for each $e,e'\in{\mathcal{E}_{train}}$ is shared across these works.
3. The paper could benefit from some additions like a mathematical result demonstrating that the approach achieves some well-motivated desirable property on the linear-gaussian setting. Another interesting aspect could be to discuss the causal version of the "unsupervised" graph in Fig. 1, i.e. $Z_e \leftarrow X \rightarrow Z_{inv}$ and derive the corresponding architecture.
4. Finally, a clear drawback is that experiments are performed in rather small datasets and carefully designed problems that follow the assumptions of the method.

Small comments: It might be worthwhile enlarging Figure 1 and explaining the terms FIIF and PIIF formally. There are several places that use terms like "recover the causal structure" (line 248), "causality constraints" etc. I think it's better to keep the word invariance rather than causality, because recovering the causal structure might allude some readers to think the work tried to do causal discovery or some sort of structure learning, which is not the case.

Overall, while I have several reservations about the paper, I gave an overall score of borderline accept and will reconsider it upon the authors' response.

[1] Bousmalis, Konstantinos, et al. "Domain separation networks." Advances in neural information processing systems 29 (2016).
[2] Muandet, Krikamol, David Balduzzi, and Bernhard Schölkopf. "Domain generalization via invariant feature representation." International conference on machine learning. PMLR, 2013.
[3] Sun, Baochen, Jiashi Feng, and Kate Saenko. "Correlation alignment for unsupervised domain adaptation." Domain adaptation in computer vision applications. Cham: Springer International Publishing, 2017. 153-171.
[4] Sicilia, Anthony, Xingchen Zhao, and Seong Jae Hwang. "Domain adversarial neural networks for domain generalization: When it works and how to improve." Machine Learning 112.7 (2023): 2685-2721.

**Questions:**

What is the conceptual difference from methods mentioned in the "weaknesses" part? The presence of a label is a technical difference, but the motivation for replacing the label with a reconstruction loss seems somewhat weak.

It is mentioned that a separate encoder is trained for each environment. Is it actually a separate set of weights being trained, and if so then why? It seems more natural to train an encode that takes a one hot encoding of the environment, and will leverage the larger combined dataset to train its weights, while also possible learning similarities between the domains.

---

> ### Author Response · Authors · 2025-11-19
>
> Hello,
>
> We thank the reviewer for the valuable input on our paper.
>
> We would first like to address the weaknesses you mentioned:
>
> 1. Since Arjovsky et al. wrote their seminal paper on IRM in 2019 until today, this problem has been dealt with and seen through the lens of supervised learning. Our work tries to widen the scope and introduce an unsupervised framework to deal with distribution shifts between environments. We believe that this framing is very helpful and enables advances in problems where labels are scares and/or expensive, and also for tasks which are fundamentally unsupervised. Some examples one can think of:
>
>   ● Collecting data for autonomous driving tasks during a sunny day, and using the UIRM framework to generalize for the task of driving during nighttime and/or heavy rainfall.
>
>   ● Collecting MRI samples (which are expensive to label) from several different hospitals (each hospital can be treated as a different environment) and preprocessing them using UIRM so that there are no spurious features that depend on the equipment of each specific hospital.
>
> To summarize, our goal in UIRM is to separate between the environment-invariant features to the environment-dependent features. Having this in mind, we can now dive into section 4.3 (we assume that you meant section 4.3- “Back to Supervised Learning”). In this section, we try to demonstrate that our encoders manage to capture the information that they are intended to, while the invariant encoder successfully avoids obtaining information regarding the environment, which is a key desirable trait for the embedding in traditional supervised IRM.
> We believe the name we gave this work is suitable because of the link to the line of work already done in the IRM domain, and the consideration of our framework in comparison to it. There have also been a few works which used the term “risk” for unsupervised learning objectives such as:
>
>   ● “Unsupervised Learning Without Overfitting: Empirical Risk Approximation as an Induction Principle for Reliable Clustering” by JM Buhmann, M Held‏
>
>   ● “Robust Unsupervised Learning via L-Statistic Minimization” by A Maurer, DA Parletta, A Paudice, M Pontil
>
> Having said that, we are open to reconsidering the title of the paper if you do not find our arguments satisfying.
>
> 2. We thank you for bringing these important papers to our attention, we have addressed them in our current paper revision (section 2, starting at line 154). Detailed response regarding each paper can be found below, as an answer to your concrete questions.
>
> 3. Further analysis of the linear Gaussian settings can be found in section A in the appendix.
> As for the case in which the causality direction is reversed, we did not encounter this approach in the IRM literature. The data generating process is often modeled with the latent variables containing the sematic information, which in turn determines what “appears” in $X$. However, we can investigate that if you think this direction shows promise.
>
> 4. We hope that the CelebA experiments can be considered as an interesting demonstration and direction, and can provide motivation for more practical future use cases for our method.
>
> Regarding the small comment, we have taken it into account and incorporated it into the paper’s revision. We provided a citation with further explanation to PIIF and FIIF (line 127) for more context, and removed the word “causal” from the unnecessary places you mentioned.
>
> We will address your questions in the next comment due to limited space.

---

> > ### Author Response · Authors · 2025-11-19
> > **Questions Answering**
> >
> > We would now like to move on and address your questions:
> >
> > 1. Let us first examine our paper against each of the papers you have provided and highlight our contribution in comparison:
> >
> >   ● DSN is a supervised method (labels are needed), also, no underlying SCM is assumed and therefore causality constraints cannot be exploited for the derivation of the algorithm. Finally, no generative framework is provided and therefore, no environment-controlled sampling and/or environment transfer can be performed.
> >
> >   ● DICA (or more precisely UDICA for the unsupervised settings) focuses on maximizing the variance, which is less general than our likelihood objective. It also does not use assumptions on the underlying SCM or provides generative capabilities.
> >
> >   ● CORAL has similar “limitations” as DICA for that matter.
> >
> >   ● DANN also focuses on the supervised framework and has similar “limitations” as the papers mentioned above.
> >
> > Overall, the lack of labels is a major difference in our opinion, and is also what separates our work from the rich literature and past work on IRM. Another key difference is that these works do not consider the specific framework of IRM, and therefore, their solutions are inherently different and do not consider the same assumptions as ours. In summary, our work offers an unsupervised framework to solve the IRM problem, we provide two novel algorithms within this framework, and offer the “environment transfer” scheme for alleviating distribution shifts between environments. This in turn leads to a solution that is also applicable to the realm of domain adaptation/ generalization.
> >
> > 2. The existence of a separate encoder allows the model to learn a different probability distribution for each environment, which in turn makes it possible to generate samples from different environments (as we have different priors for each environment) and to perform environment transfer.
> >
> > Thank you again for the thorough review and important remarks and clarifications, we hope we have addressed your concerns in a satisfactory manner. We will be happy to address any further questions and concerns you might have.

---

> > > ### Comment · Reviewer_vC27 · 2025-11-19
> > > **Post-rebuttal update**
> > >
> > > Thank you for the response.
> > >
> > > About the name and distinguishing factors from other works:
> > >
> > > In the context of enforcing independence relations, both the original ICP of Peters et al. and the representation learning variant of IRM specifically discuss the invariance of mechanisms, i.e. relations between a subset of features, or a function of features, and a variable Y. So the words “prediction” and “risk minimzation” for both papers respectively, are rather helpful in distinguishing the conditional independence roughly given by $Y \bot E | \phi(X)$ studied in these papers, from $\phi(X) \bot E$ which is the independence studied in the paper under review (sometimes called domain invariance, invariant features, or other names in the papers I referenced and others). The differences between these independence relations, and when each one is suitable, is also discussed in several works. I think that calling your work Unsupervised IRM adds more confusion to literature that is already quite difficult to navigate because terms are abused.
> > >
> > > Based on the response to my question about related work, it also seems like the main contribution of this work is methodological (i.e. using an autoencoder). Prior work considered the problems of learning invariant features, but did this by maximizing variance instead of using an autoencoder, so this is another reason to reconsider the title. I’m not sure why considering the specific framework of IRM is helpful here, or how the specific framework of IRM that’s discussed in the rebuttal different from previous works such as Muandet et al. and others. It seems like the motivation is to guarantee robust prediction on a new environment, but I don’t think the mathematical results in the paper show that specifically an autoencoder is somehow helpful for that.
> > >
> > > > As for the case in which the causality direction is reversed, we did not encounter this approach in the IRM literature. The data generating process is often modeled with the latent variables containing the sematic information, which in turn determines what “appears” in X.
> > >
> > > For papers that study causal and anti-causal causal problems, perhaps the most relevant paper is [1]. There are other works (e.g. [2]) that come to mind. This is beyond the scope of an edit to this submission, but could be of interest if you wish to expand this work beyond the contribution of the autoencoder architecture alone.
> > >
> > > I will keep my score for now and wait to hear the opinions of other reviewers and the AC before finalizing it.
> > >
> > > [1] Veitch et al. Counterfactual Invariance to Spurious Correlations in Text Classification. NeurIPS 21
> > >
> > > [2] Feder et al. In the eye of the beholder: Robust prediction with causal user modeling. NeurIPS 22

---

> > > > ### Author Response · Authors · 2025-11-24
> > > >
> > > > Thank you for the follow-up.
> > > >
> > > > We accept the points you make. Because our paper relies and builds mostly on the literature in IRM, we suggest modifying the title to something along the lines of “Unsupervised Representation Learning - an Invariant Risk Minimization perspective”.
> > > > We hope you find this option (or something similar) more suitable and less confusing.
> > > >
> > > > As for the papers regarding the anti-causal case, we thank you for bringing them to our attention, we’ll consider this direction for future work.

---

> ### Author Response · Authors · 2025-12-03
>
> To summarize, we believe it is important to emphasize that a key novelty of our work is introducing unsupervised representation learning into the IRM setup. We show mathematically that PICA recovers only invariant directions (Section A.2) under linearity and Gaussian assumptions. For VIAE, we provide empirical qualitative evidence demonstrating its performance in Sections 4 and D. In addition, Section 4.3 examines the separation the autoencoder achieves between environmental and invariant features.
>
> Finally, we have updated the title of the paper to avoid confusion.
>
> We thank the reviewer again for the feedback and insightful comments, and hope we have managed to address the raised concerns.

---

### Meta-Review · Area_Chair_7ZxN · 2026-01-12

**Summary:**

There is no consensus between reviewers.

While I tend to agree with `XTfo`, that the VIAE is an interesting structure, I also agree with `wMwF` that older pieces of the literature should be incorporated into the discussion. There are variational auto-encoders with invariance properties matching the described one as far as I can tell (Lopez et al 2018, Moyer et al 2018), and the extensive disentanglement literature as cited by `wMwF`. While we understand that the present work does not attempt within-group disentanglement, surely the separation of sources into environment dependent and environment invariant features relates to this work.

I have recommended acceptance, but I encourage the authors to continue to revise their work, since all 3 reviewers have some conceptual or notation confusion.

Lopez, Romain, et al. "Information constraints on auto-encoding variational bayes." Advances in neural information processing systems 31 (2018).

Moyer, Daniel, et al. "Invariant representations without adversarial training." Advances in neural information processing systems 31 (2018).

**Reviewer Concerns:**

See previous.

**Reviewer Scores:**

None.

---

### Decision · Program_Chairs · 2026-01-26

Accept (Poster)